# Single molecule counting detects low-copy glycine receptors in hippocampal and striatal synapses

**Serena Camuso[1], Yana Vella[2], Souad Youjil Abadi[1], Clémence Mille[1], Bert Brône[2], Christian G Specht[1]***

[1]Neuro-Bicêtre, Inserm U1195, Université Paris-Saclay, Paris, France; [2]UHasselt, Neurophysiology Laboratory, BIOMED Research Institute, Hasselt, Belgium

## eLife Assessment

The study presents **convincing** quantitative evidence, supported by appropriate negative controls, for the presence of low-abundance glycine receptors (GlyRs) within inhibitory synapses in telencephalic regions of the mouse brain. Using sensitive single-molecule localization microscopy of endogenously tagged GlyRs, the authors reveal previously undetected populations of these receptors. Although the functional significance of these low-abundance GlyRs remains to be established, the findings offer **valuable** insights and methodologies that will be of interest to neuroscientists studying inhibitory synapse biology.
[Editors' note: this paper was reviewed by Review Commons.]

**\*For correspondence:**
christian.specht@inserm.fr

**Abstract** Glycine receptors (GlyRs) are heteropentameric chloride channels that mediate fast inhibitory neurotransmission in the brainstem and spinal cord, where they regulate motor and sensory processes. GlyRs are clustered in the postsynaptic membrane by strong interactions of the β subunit with the scaffold protein gephyrin. Even though *Glrb* mRNA is highly expressed throughout the brain, the existence of synaptic GlyRs remains controversial as there is little conclusive evidence using conventional fluorescence microscopy and electrophysiological recordings. Here, we exploit the high sensitivity and spatial resolution of single molecule localisation microscopy (SMLM) to investigate the presence of GlyRs at inhibitory synapses in the brain, focusing on several areas of the telencephalon. Making use of a knock-in mouse model expressing endogenous mEos4b-tagged GlyRβ, we identified few GlyRs in sub-regions of the hippocampus. Dual-colour SMLM revealed that these sparse receptors are integrated within the postsynaptic gephyrin domain, pointing to a possible role in maintaining the structural integrity of inhibitory synapses. In contrast, we found functionally relevant numbers of synaptic GlyRs at inhibitory synapses in the ventral striatum. Our results highlight the strength of SMLM to detect few and sparsely distributed synaptic molecules in complex samples and to analyse their organisation with high spatial precision.

## Introduction

Inhibitory neurotransmission in the central nervous system (CNS) is largely mediated by glycine receptors (GlyRs) and γ-aminobutyric acid type A receptors (GABA$_A$Rs) (*Alvarez, 2017*; *Kasaragod and Schindelin, 2018*). Both classes of receptors are widely expressed throughout the brain and the spinal cord; however, they have specific regional expression patterns. GlyRs are particularly abundant in the spinal cord and brainstem, where they play an important role in the processing of sensory and motor information, as well as the modulation of pain responses (*Alvarez, 2017*; *Fenech et al.,*

2024). Although the expression of GlyRs is much lower in the brain (*Maynard et al., 2021*; *Zeilhofer et al., 2005*), glycinergic transmission is known to be involved in reward signalling and possibly pain-associated responses (*Adermark et al., 2011*; *Devoght et al., 2023*; *Fenech et al., 2024*; *Muñoz et al., 2018*; *San Martin et al., 2020*).

Both GABA$_A$Rs and GlyRs are cys-loop pentameric chloride channels that are composed of different combinations of subunits. Each subunit contains a large N-terminal extracellular domain, four trans-membrane domains (TM1–4), as well as a flexible intracellular domain (ICD) between TM3 and TM4 (*Kasaragod and Schindelin, 2018*). Pentameric GlyRs are assembled from five different subunits, α1-α4 and β. Homopentameric GlyRs composed only of α subunits are mostly found in the extrasynaptic plasma membrane. In contrast, heteropentameric receptors containing both α and β subunits accumulate at postsynaptic sites, due to a direct interaction between the ICD of the β subunits and gephyrin, the main scaffold protein at inhibitory synapses (*Alvarez, 2017*).

In situ hybridisation studies have shown that the mRNA of the GlyRα1 and β subunits is highly expressed in spinal cord neurons (*Ceder et al., 2024*; *Malosio et al., 1991*). The *Glrb* transcript is also expressed in most brain regions, including olfactory bulb, cerebral cortex, hippocampus, and striatum (*Ceder et al., 2024*; *Fujita et al., 1991*; *Malosio et al., 1991*). Surprisingly, expression of GlyRβ protein appears to be exceedingly low in the telencephalon. For example, GlyR labelling was only detected at a few synapses in the hippocampus, mainly in the pyramidal layer (*Danglot et al., 2004*; *Maynard et al., 2021*; *Weltzien et al., 2012*). In line with this, electrophysiological measurements have failed to detect synaptic GlyR currents in the hippocampus (*Chattipakorn and McMahon, 2002*; *Mori et al., 2002*; *Song et al., 2006*), with only a single study reporting evoked inhibitory postsynaptic currents (IPSCs) in mouse hippocampal CA1 pyramidal cells (*Muller et al., 2013*).

The mRNA levels of the GlyR α subunits are generally low across the brain (*Ceder et al., 2024*; *Malosio et al., 1991*). Yet, surface expression of homopentameric GlyRs was demonstrated by electrophysiological recordings of glycine-induced currents in several regions, including the hippocampus and dorsal striatum (*Chattipakorn and McMahon, 2002*; *Comhair et al., 2018*; *Molchanova et al., 2017*; *Mori et al., 2002*; *Song et al., 2006*). These extrasynaptic receptors are thought to play a role in the tonic inhibition of central neurons (*McCracken et al., 2017*; *Mori et al., 2002*; *Song et al., 2006*). This situation appears to be different in the ventral striatum (*nucleus accumbens*), where glycinergic miniature inhibitory postsynaptic currents (mIPSCs) likely corresponding to heteropentameric GlyRs have been reported (*Muñoz et al., 2018*), in addition to homopentameric receptors (*McCracken et al., 2017*; *Molander and Söderpalm, 2005*; *Muñoz et al., 2020*).

To explore the presence of GlyRs at synapses in the brain, we made use of the high spatial resolution and extraordinary sensitivity of single molecule localisation microscopy (SMLM), which enabled us to detect individual GlyR complexes in different regions of the telencephalon, including the hippocampal formation and striatum. Using a knock-in (KI) mouse model that expresses endogenous mEos4b-tagged GlyRβ subunits (*Maynard et al., 2021*), we identified low-copy numbers of GlyRs at inhibitory synapses. Dual-colour SMLM further demonstrated the integration of these GlyRs within the postsynaptic gephyrin scaffold, suggesting that they are important for the assembly or maintenance of inhibitory synaptic structures in the brain.

## Results

### Identification of low-copy synaptic GlyRs in mouse hippocampus

The distribution of glycinergic synapses is mainly confined to the spinal cord and brainstem (*Alvarez, 2017*). While the presence of the *Glrb* mRNA transcript is well documented in different brain areas (*Ceder et al., 2024*; *Fujita et al., 1991*; *Malosio et al., 1991*; *Figure 1*), little is known about GlyRβ protein expression in the brain (*Danglot et al., 2004*; *Maynard et al., 2021*). Here, we exploited the high sensitivity and spatial resolution of SMLM to probe the presence of GlyRβ subunits at synapses in the brain, focusing on the hippocampus.

Since no reliable antibodies against the β subunit of the GlyR are available, we used a KI mouse model expressing endogenous mEos4b-tagged GlyRβ subunits (*Maynard et al., 2021*; *Wiessler et al., 2024*). No obvious functional, behavioural, or ultrastructural phenotypes have been reported in homozygous and heterozygous animals expressing mEos4b-GlyRβ. Thin cryostat coronal sections (10 μm) were cut from homozygous *Glrb*$^{eos/eos}$ mouse brain and labelled with NeuN antibody and with

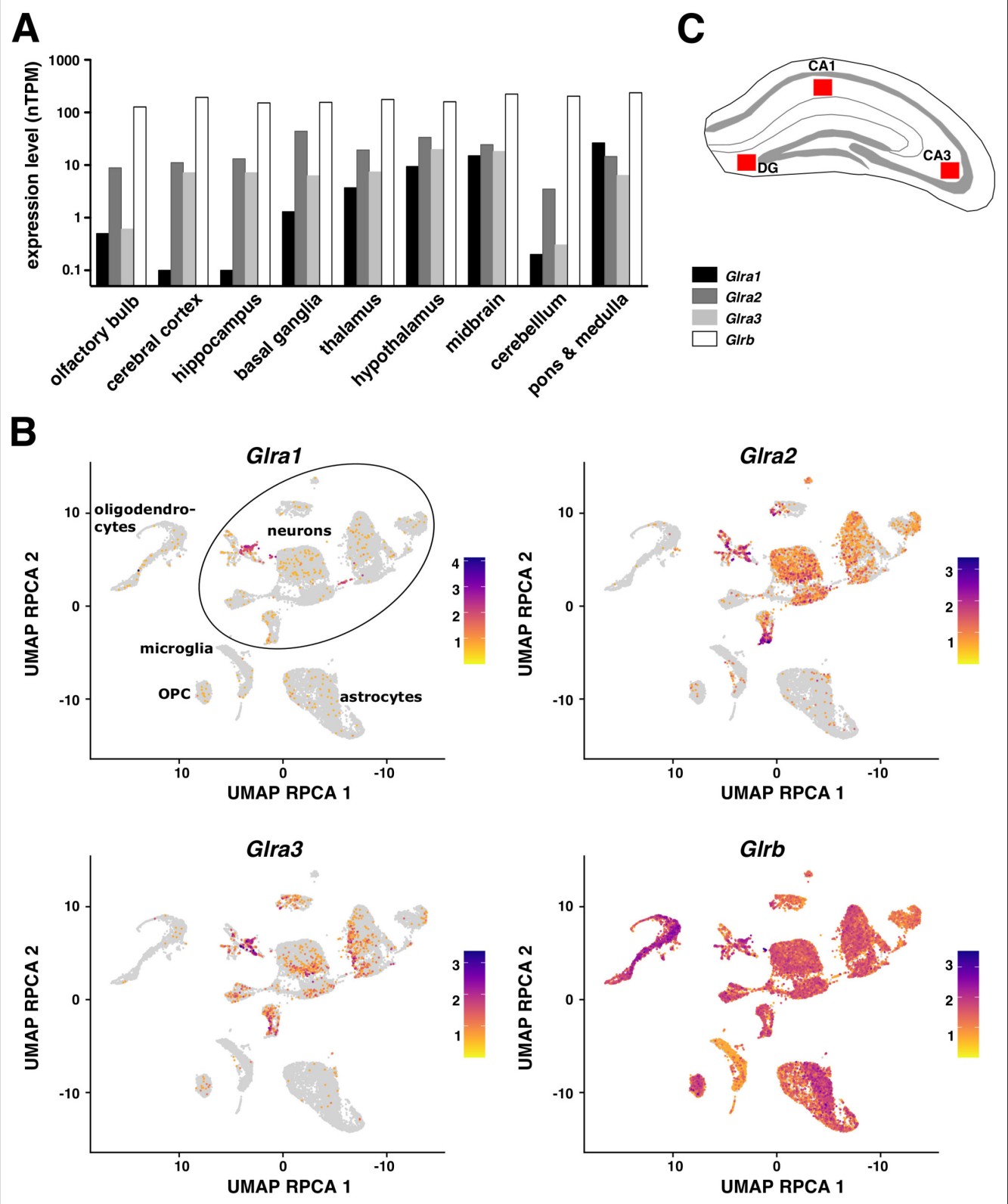

**Figure 1.** Glycine receptor (GlyR) gene expression in mouse brain. (**A**) mRNA expression of GlyR subunits α1, α2, α3, and β in different brain regions of 2-month-old C57BL/6J mice (sorted from frontal regions on the left to dorsal on the right). RNA-seq data were retrieved from Human Protein Atlas (HPA, https://www.proteinatlas.org/) and are expressed as normalised transcripts per million (nTPM). The hippocampus includes the subregions of the *cornu ammonis* (CA) and the dentate gyrus (DG). (**B**) UMAP projection (using RPCA reduction) of single-cell transcriptomic data derived from four mouse

*Figure 1 continued on next page*

*Figure 1 continued*

brain regions: hippocampus (HIP-CA), dorsal striatum (STRd), ventral striatum (STRv), and medulla (see Methods for details on data acquisition and processing). Each panel shows the expression of a different GlyR subunit (*Glra1*, *Glra2*, *Glra3*, or *Glrb*), visualised with a colour scale from low (yellow) to high (purple). Clusters corresponding to major cell types (neurons, oligodendrocytes, oligodendrocyte precursor cells [OPCs], astrocytes, and microglia) are labelled in the first panel. (**C**) Schematic diagram of hippocampus. The red squares represent the hippocampal sub-regions in which SMLM recordings were taken; the molecular layer of the DG and the *stratum radiatum* of CA3 and CA1.

Sylite, a small peptide probe against the inhibitory synaptic scaffold protein gephyrin (*Khayenko et al., 2022*). The analyses were conducted in the molecular layer of the dentate gyrus (DG), where the dendrites of granule cells receive synaptic inputs from the entorhinal cortex, and in the *stratum radiatum* of the CA3 and CA1 regions, where the apical dendrites of pyramidal cells make contact with Mossy fibres and Schaffer collaterals, respectively (*Figure 1C*). The different hippocampal sub-regions were identified using the neuronal marker NeuN in the green channel (not shown). Reference images

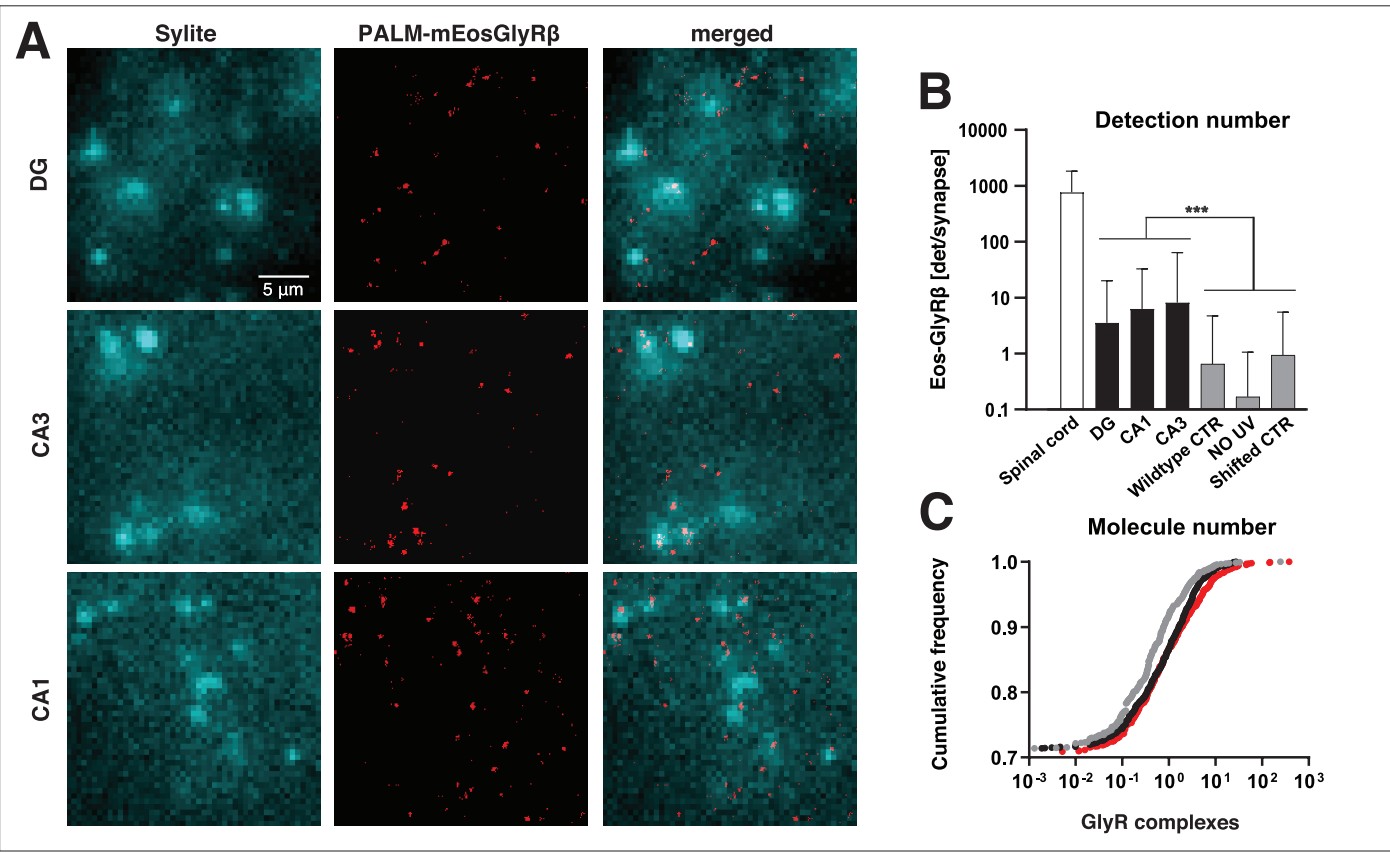

**Figure 2.** Single molecule localisation microscopy (SMLM) of mEos4b-GlyRβ subunits in the hippocampus. (**A**) Single molecule detections of mEos4b-GlyRβ (red SMLM pointillist image) in the dentate gyrus (DG), CA1 and CA3 regions of the hippocampus in 10 µm cryostat sections of the knock-in mouse line *Glrb*^eos/eos at postnatal day 40 (red pointillist images). Inhibitory synapses were identified in epifluorescence images using the gephyrin marker Sylite (cyan). Scale bar: 5 µm. (**B**) Mean number of mEos4b-GlyRβ detections per synaptic gephyrin cluster in spinal cord (n = 1265 synaptic clusters from 8 fields of view) and brain slices (n > 5000 clusters from 9 fields of view for each region) from N = 3 independent experiments corresponding to 3 *Glrb*^eos/eos mice (N = 2 for spinal cord). Recordings were also made in the CA3 of wildtype mice not expressing endogenous mEos4b-GlyRβ (negative control; n = 1424 clusters, 3 fields of view, 2 *Glrb*^WT/WT mice) and in *Glrb*^eos/eos hippocampal slices without photoconversion of mEos4b (no UV, n = 1214 clusters, 9 fields of view, 3 animals). A pixel shift control with flipped images was done with the same dataset (*Glrb*^eos/eos, CA3; n = 1978 clusters, 9 fields of view, 3 animals). Data are shown as mean ± SD. Levels of significance were determined using a Kruskal-Wallis test with Dunn's multiple comparison test: ***p < 0.0001. (**C**) Cumulative distribution representing the estimated copy number of mEos4b-GlyRβ complexes per synapse in DG (grey line), CA1 (black line), and CA3 (red line) regions. Copy numbers were background-corrected by subtracting the value obtained for the negative control in wildtype slices.

The online version of this article includes the following figure supplement(s) for figure 2:

**Figure supplement 1.** Single molecule localisation microscopy (SMLM) of endogenous mEos4b-GlyRβ in spinal cord.

of Sylite were taken in the far-red channel, followed by SMLM recordings of the green-to-red photo-convertible fluorescent protein mEos4b attached to the GlyRβ subunit (*Figure 2A*).

We observed very few single molecule detections during SMLM, indicating exceedingly low mEos4b-GlyRβ expression in the hippocampus. SMLM super-resolution images were reconstructed and analysed using Icy software to quantify the number of mEos4b detections per gephyrin cluster (*Figure 2B*). We counted on average between 3 and 10 detections per synapse in all hippocampal sub-regions. This is about two orders of magnitude lower than in the spinal cord (p < 0.0001, non-parametric Kruskal-Wallis [KW] one-way ANOVA with Dunn's multiple comparison test, KW test), where we counted almost 1000 detections of mEos4b-GlyRβ per synapse (*Figure 2B*, *Figure 2—figure supplement 1*). Aside from the low detection numbers in the hippocampus, we did not find obvious differences between the sub-regions, with a significant difference observed only between the DG and CA1 (p = 0.0002, KW test).

To ascertain that what we saw in the hippocampus were indeed mEos4b detections and not imaging artefacts, we performed control experiments in hippocampal slices from wildtype mice not expressing mEos4b-tagged GlyRβ subunits. As expected, the mean number of detections at CA3 synapses was significantly lower in the negative control than in the *Glrb*[eos/eos] slices (*Figure 2B*; p < 0.0001, KW test). Similarly, we recorded SMLM movies in *Glrb*[eos/eos] slices without 405 nm illumination. In the absence of UV, the mEos4b fluorescent protein is not converted into its red form, making this a stringent internal control. The number of detections per gephyrin cluster was again much lower than in the recordings with 405 nm laser illumination (*Figure 2B*; p < 0.0001, KW). These data confirm that the non-specific background of detections, mainly outside of synapses, is sufficiently low to accurately identify and quantify endogenous mEos4b-GlyRβ subunits in the hippocampus of our KI mouse model. To exclude the possibility that the co-localisation between the mEos4b-GlyRβ detections and gephyrin was due to a random overlap, we did a pixel shift analysis with two-colour images of the CA3 region, in which the Sylite channel was horizontally flipped relative to the SMLM image of the mEos4b detections. This transformation led to a significant reduction in the number of mEos4b-GlyRβ detections at gephyrin clusters compared to the original data (*Figure 2B*; p < 0.0001, KW), confirming that the localisation of mEos4b-GlyRβ at inhibitory synapses is not down to chance.

We also estimated the absolute number of GlyRs per synapse in the hippocampus. The number of mEos4b detections was converted into copy numbers by dividing the detections at synapses by the average number of detections of individual mEos4b-GlyRβ containing receptors. This value was measured across CA3 slices, both at synapses and in the extrasynaptic region (see Methods). The advantage of this counting strategy is that it is independent of the stoichiometry of heteropentameric GlyRs that is still under debate (e.g. *Durisic et al., 2012*; *Grudzinska et al., 2005*; *Patrizio et al., 2017*; *Zhu and Gouaux, 2021*). The obtained copy numbers are reported as background-subtracted values (*Table 1*). According to our quantification, GlyRs are present at about a quarter of hippocampal synapses, with copy numbers most often in the single digits (*Figure 2C*, *Table 1*). Only a few synapses contain 10 or more GlyRs, and as many as 75% of hippocampal synapses do not contain any GlyRs at all. No obvious differences were noticed in the copy number of GlyRs between sub-regions. Taken together, our findings demonstrate the existence of very few heteropentameric GlyR complexes at

**Table 1.** Estimated copy numbers of mEos4b-GlyRβ containing heteropentameric glycine receptors (GlyR) complexes at inhibitory synapses in different regions of the central nervous system (CNS) of *Glrb*[eos/eos] knock-in (KI) mice (background-corrected, see Methods).

| CNS region | GlyR copy number (mean ± SEM) | Range of GlyR copies (5–95 percentile) | Fraction of GlyR-positive synapses (≥0.5 copies) |
|---|---|---|---|
| Spinal cord | 120 ± 5 | 0 – 470 | 0.89 |
| CA1 | 0.34 ± 0.05 | 0 – 3 | 0.18 |
| CA3 | 1.11 ± 0.30 | 0 – 5 | 0.18 |
| DG | 0.35 ± 0.18 | 0 – 2 | 0.13 |
| Dorsal striatum | 3.00 ± 0.18 | 0 – 15 | 0.43 |
| Ventral striatum | 26.10 ± 1.48 | 0 – 112 | 0.73 |

inhibitory synapses in the hippocampus. With an average of approximately one GlyR per synapse, this is about a hundredfold lower than in the spinal cord.

## Sub-synaptic distribution of GlyRs at hippocampal synapses

At glycinergic synapses in the spinal cord, the receptors are anchored in the postsynaptic membrane by gephyrin, a scaffold protein that binds with high affinity to the GlyRβ subunit, promoting the synaptic localisation of GlyRs (e.g. *Kasaragod and Schindelin, 2018*; *Kostrz et al., 2024*; *Maynard et al., 2021*). The nanoscale organisation of inhibitory synapses in the hippocampus containing low-copy GlyRs has not yet been studied in detail (*Danglot et al., 2004*). To obtain information about the distribution of GlyRs within the synaptic structure, we carried out dual-colour SMLM of GlyRβ and gephyrin in brain slices of adult *Glrb*^eos/eos mice using stochastic optical reconstruction microscopy (dSTORM) with organic fluorophores in a reducing buffer (*Yang and Specht, 2020*). Compared to SMLM with fluorescent proteins, dSTORM provides higher photon yields, resulting in a better localisation precision (~12 nm in our recordings, see Methods), which is critical for resolving fine structural details. For these experiments, we focused on the CA3 region of the hippocampus, where the presence of GlyRs at inhibitory synapses was demonstrated before (*Figure 2*). Due to their low number, we did not expect to see any particular organisation of the receptors. Instead, the aim was to determine whether GlyRs are actually intrinsic components of the postsynaptic domain of inhibitory hippocampal synapses.

Endogenous mEos4b-GlyRβ subunits in tissue slices of *Glrb*^eos/eos KI animals were labelled with anti-mEos-AF647 nanobodies (NanoTag). The specificity of the nanobody was tested in spinal cord slices that were also labelled with gephyrin antibody (mAb7a, Synaptic Systems) and a CF568-conjugated secondary antibody. Wide-field fluorescence images showed extensive co-localisation of all three channels (mEos4b, anti-gephyrin-CF568, anti-mEos-AF647) in *Glrb*^eos/eos slices; the non-specific background of the nanobody in wildtype animals was very low (*Figure 3—figure supplement 1*).

For dual-colour SMLM, brain slices from *Glrb*^eos/eos mice were labelled with anti-mEos-AF647 nanobody, as well as with anti-gephyrin and CF680-conjugated anti-mouse secondary antibodies (*Figure 3A*). The emitted light from the two far-red dyes was separated by spectral demixing, using a SAFe 360 nanoscope (Abbelight) equipped with a dichroic mirror at 700 nm and two sCMOS cameras for detection. In this imaging modality, the single molecule detections recorded simultaneously on the two cameras are attributed to one or the other far-red fluorophore based on their specific intensity ratio (see Methods; *Figure 3—figure supplement 2A–E*). To further test the specificity of the nanobody labelling and the demixing procedure, we performed experiments in *Glrb*^eos/eos slices in which the nanobody was omitted and only gephyrin was labelled (CF680). Under these conditions, spectral demixing produced a single peak of intensity ratios corresponding to the CF680 fluorophore, with only a few wrongly attributed detections in the AF647 channel (*Figure 3—figure supplement 2F and G*).

In line with our SMLM experiments with the photoconvertible fluorescent protein mEos4b in brain slices (*Figure 2*), we observed sparse single molecule detections of anti-mEos-AF647 nanobody that generally co-localised with dense clusters of gephyrin detections, confirming the presence of mEos4b-GlyRβ at inhibitory synapses in the hippocampus (*Figure 3A*). The two-colour SMLM pointillist images were analysed using the DBSCAN clustering tool in the NEO software to measure the distance between mEos4b-GlyRβ and the corresponding gephyrin clusters. According to our measurements, the Euclidean distance between the centre of mass (CM) of GlyRβ and gephyrin was $79 \pm 64$ nm (mean ± SD, n = 212 pairs of clusters), pointing to a close spatial relationship at inhibitory synapses (*Figure 3B*). We also calculated the relative distance of the GlyRβ detections from the CM of gephyrin, compared to the mean distance of the gephyrin detections from their own CM as defined by the radius of gyration (RG) of the gephyrin clusters. This ratio was below one for most clusters, indicating that the GlyRs are well integrated within the postsynaptic gephyrin domains at CA3 synapses (*Figure 3C*).

## Differential expression of endogenous mEos4b-GlyRβ in dorsal and ventral striatum

Several reports have described the presence of glycinergic currents in the striatum (*Molchanova et al., 2017*; *Muñoz et al., 2018*). However, the localisation of GlyRs at synapses in this brain area remains controversial. Homopentameric GlyRα2 complexes are thought to mediate tonic inhibition

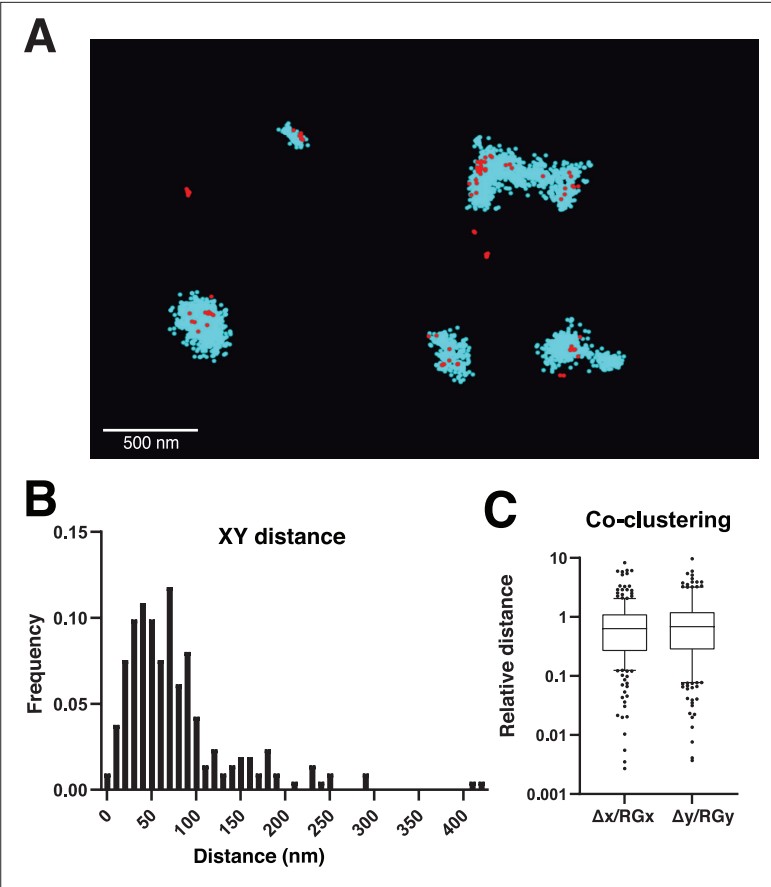

**Figure 3.** Dual-colour single molecule localisation microscopy (SMLM) of glycine receptor (GlyR) and gephyrin at inhibitory hippocampal synapses. (**A**) Dual-colour SMLM using spectral demixing of endogenous mEos4b-GlyRβ labelled with anti-mEos-AF647 nanobody (red, NanoTag), and mouse anti-gephyrin (mAb7a, Synaptic Systems) and CF680-conjugated secondary anti-mouse antibodies (cyan) in hippocampal slices of the *Glrb*[eos/eos] knock-in mouse line at postnatal day 40. Scale bar: 500 nm. (**B**) Euclidean distance between the centre of mass (CM) of the anti-mEos-AF647 nanobody (GlyRβ) and gephyrin (mAb7a-CF680) detections of corresponding clusters. (**C**) Distance of the CM of the anti-mEos-AF647 detections from the CM of gephyrin relative to the radius of gyration (RG) of the gephyrin cluster along the x and y axes (Δx/RGx, Δy/RGy). N = 2 independent experiments corresponding to 2 *Glrb*[eos/eos] animals.

The online version of this article includes the following figure supplement(s) for figure 3:

**Figure supplement 1.** Specificity of the anti-mEos-AF647 nanobody.

**Figure supplement 2.** Dual-colour single molecule localisation microscopy (SMLM) using spectral demixing.

in the dorsal striatum (***Devoght et al., 2023***; ***Molchanova et al., 2017***), yet glycinergic mIPSCs were detected in the *nucleus accumbens*, i.e., part of the ventral striatum (***Muñoz et al., 2018***). To resolve this issue, we quantified the GlyRβ-containing synaptic receptors in the striatum of *Glrb*[eos/eos] animals using SMLM.

The number of detections of mEos4b-GlyRβ per gephyrin cluster, as well as the resulting copy number of GlyRs at synapses, was measured in the dorsal and in the ventral striatum in *Glrb*[eos/eos] mice at postnatal day 40. Coronal cryo-sections of 10 μm thickness were immunolabelled for NeuN to distinguish the two sub-regions of the striatum. Gephyrin was labelled with the Sylite probe in order to identify inhibitory synapses (***Figure 4A***). The average number of mEos4b-GlyRβ detections per gephyrin cluster was significantly higher in the ventral striatum compared to the dorsal striatum (136 ± 5 vs 25 ± 1 detections per synapse, mean ± SEM, n = 2985 and 2719 synapses, respectively; p < 0.0001, non-parametric Mann-Whitney test [MW] test; ***Figure 4B***). Conversion of the detection numbers into receptor copy numbers revealed that about 40% of synapses in the ventral striatum contain at least 10 heteropentameric GlyR complexes, many more than in the dorsal striatum (p < 0.0001, KW

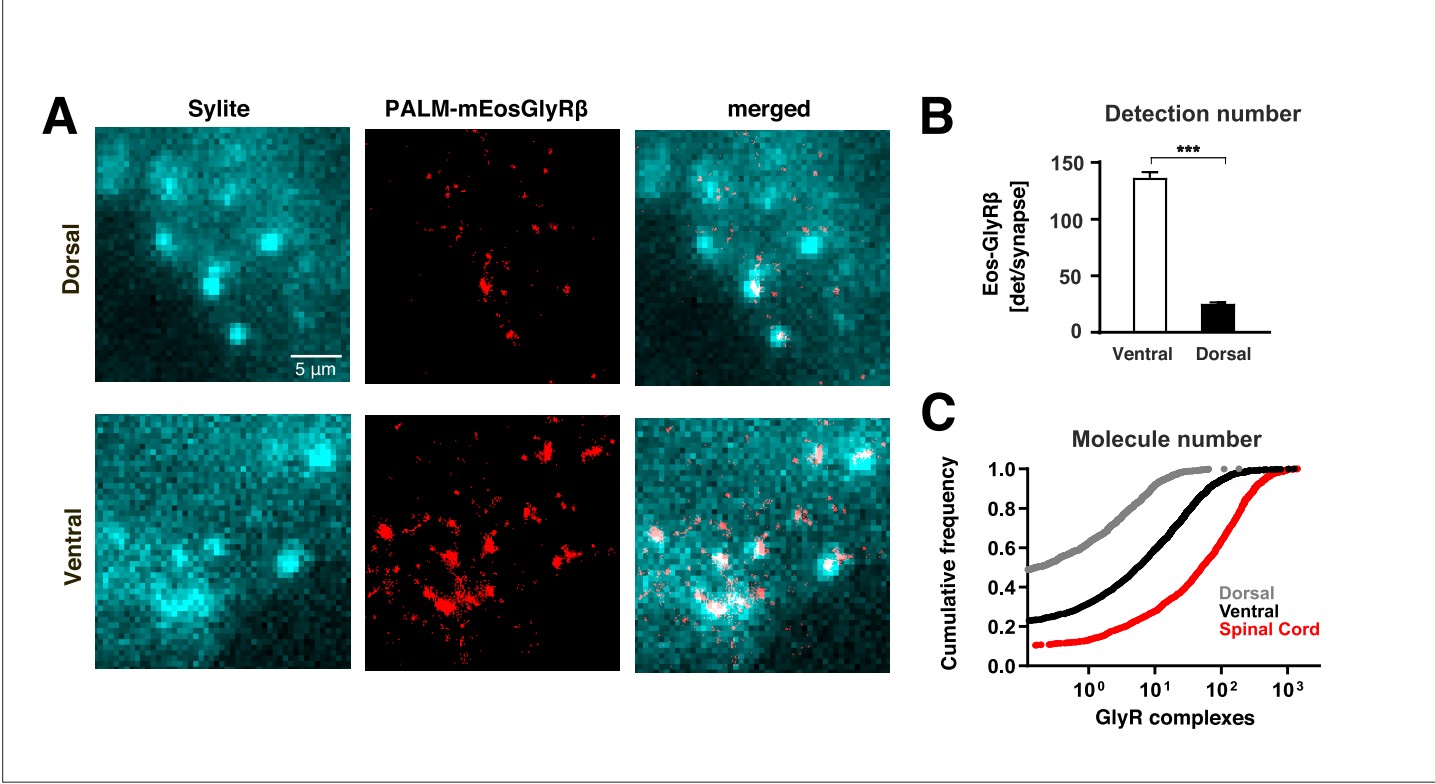

**Figure 4.** Quantitative single molecule localisation microscopy (SMLM) of endogenous glycine receptors (GlyRs) at synapses in the striatum. (**A**) Super-resolution imaging of mEos4b-GlyRβ subunits (in red) in the dorsal and ventral striatum of *Glrb*[eos/eos] knock-in mice. Cryostat slices were labelled with the gephyrin marker Sylite (cyan). Scale bar: 5 μm. (**B**) Single molecule detection numbers of mEos4b-GlyRβ per gephyrin cluster (n=2985 and 2719 clusters for ventral and dorsal striatum, respectively, from 6 fields of view per sub-region and N = 3 independent experiments from three animals; mean ± SEM; two-tailed Mann-Whitney test: ***p < 0.0001). (**C**) Cumulative distribution of the estimated number of mEos4b-GlyRβ containing receptor complexes per synapse in the dorsal striatum (grey line), ventral striatum (black line), and spinal cord (red line). Copy numbers are background-corrected (see *Table 1*).

test; *Figure 4C*, *Table 1*). As before, copy numbers are given as background-corrected values (see Methods). The same quantification was done in the spinal cord, where we found significantly higher copy numbers of receptor complexes containing mEos4b-GlyRβ (p < 0.0001 against both striatal regions, KW). With an average of 120 ± 5 receptor complexes per synaptic cluster (mean ± SEM, n = 1265 synapses, *Table 1*), these values are close to previous estimates (*Maynard et al., 2021*).

Our findings show that the numbers of mEos4b-GlyRβ subunits are much higher at synapses in the striatum compared to synapses in the hippocampal formation, and that their distribution in the striatum is area-specific. This observation is supported by electrophysiological recordings of glycinergic synaptic currents in the *nucleus accumbens* (*Muñoz et al., 2018*) as opposed to mainly extrasynaptic tonic currents in the dorsal striatum (*Molchanova et al., 2017*).

## Miniature synaptic currents in the ventral striatum recorded with whole-cell patch clamp

Since the published recordings of glycinergic currents in the striatum were performed under different conditions, i.e., different striatal sub-regions, mouse strains, and ages (*Molchanova et al., 2017*; *Muñoz et al., 2018*), we wanted to verify the area-specific functional differences in a single experimental setting. We pharmacologically isolated glycinergic mIPSCs and determined their frequency and amplitudes in brain slices of both ventral and dorsal striatum of C57BL/6J mice at postnatal days 35–41 using whole-cell patch clamp (*Figure 5*). mIPSCs are considered a measure of functional synapses because they are generated by the spontaneous release of synaptic vesicles from presynaptic terminals. Our data confirm the presence of glycinergic mIPSCs in medium spiny neurons (MSNs) in the ventral striatum and the near-complete absence of glycinergic mIPSCs in dorsal striatal MSNs (p < 0.05, MW test; *Figure 5D*). With an amplitude of 17.2 ± 2.9 pA (mean ± SEM, n = 5 cells from

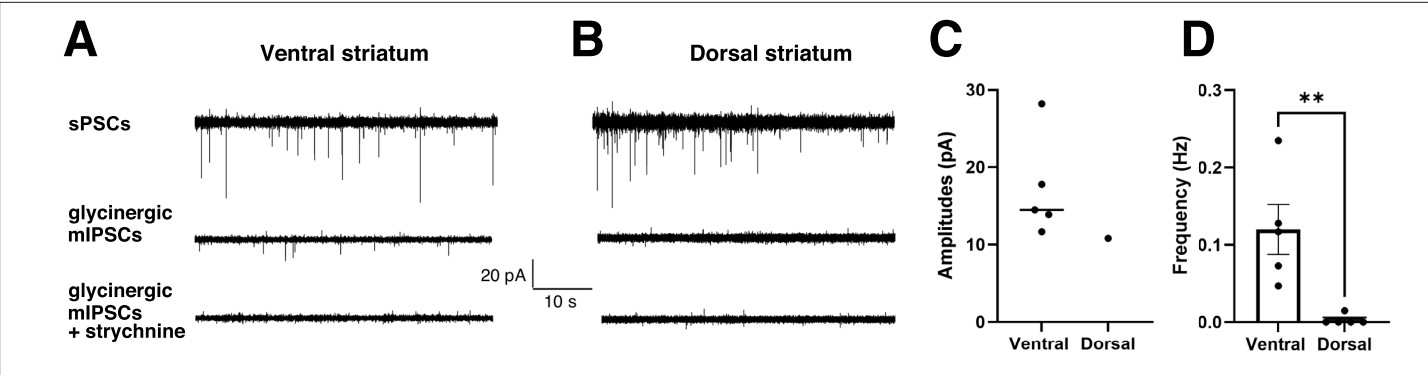

**Figure 5.** Glycinergic miniature inhibitory postsynaptic currents (mIPSCs) of medium spiny neurons (MSNs) in ventral but not in dorsal striatum. (**A–B**) Representative ventral and dorsal current traces recorded using whole-cell patch clamp in MSNs in the ventral (**A**) and dorsal striatum (**B**). The top traces show spontaneous postsynaptic current (sPSCs) recorded during aCSF application to confirm whole-cell recording. The middle traces show the pharmacologically isolated glycinergic mIPSCs during the application of aCSF containing blockers (10 µM DNQX, 0.1 µM DHBE, 5 µM L-689560, 0.5 µM tetradotoxin, and 10 µM bicuculline) present in the ventral striatum and absent in dorsal striatum. Blocking the mIPSCs by 1 µM strychnine confirms their glycinergic identity. (**C**) Quantification of the amplitude and (**D**) frequency of glycinergic mIPSCs in ventral and dorsal MSNs (mean ± SEM; n = 5 cells from 3 animals in ventral striatum and n = 5 cells from 4 animals in dorsal striatum). Levels of significance determined using a one-tailed Mann-Whitney test (**$p < 0.05$).

N = 3 animals; *Figure 5C*) and a frequency of 0.12 ± 0.03 Hz (*Figure 5D*), the mIPSC recordings in the ventral striatum are similar to the results of an earlier study of the *nucleus accumbens* (*Muñoz et al., 2018*). Our electrophysiological data also match the differences in synaptic GlyR copy numbers obtained by quantitative SMLM (*Figure 4C*).

## Expression of recombinant GlyR subunits in cultured hippocampal neurons

Synaptic GlyRs are heteropentamers composed of α (α1–α4) and β subunits, and their assembly and synaptic targeting requires the presence of both types of subunit (*Alvarez, 2017*). Since the β transcript appears to be highly expressed in most neurons, including in the hippocampus (*Figure 1*), we hypothesised that the expression of α subunits could be the limiting factor controlling the number of synaptic GlyRs. We therefore expressed different recombinant GlyR subunits in cultured hippocampal neurons and analysed their accumulation at inhibitory synapses.

Cells were transduced with lentivirus expressing mEos4b-tagged GlyR α1, α2, or β at day in vitro 3 (DIV3), fixed at DIV16, and stained with Sylite to identify synaptic gephyrin clusters. We observed co-localisation of mEos4b and Sylite puncta in neurons infected with either of the virus constructs, indicating that some cultured hippocampal neurons express the corresponding endogenous α or β subunits (*Figure 6A*). However, synaptic GlyR puncta were not found in all neurons despite using high virus titres. This means that low mRNA levels of GlyR α and/or β subunits in hippocampal neurons can limit the expression of the heteropentameric receptors. Interestingly, the mEos4b-GlyRα1 construct was expressed at a higher rate than the other subunits and had a punctate appearance in most neurons (*Figure 6A*). The majority of these puncta co-localised with gephyrin, implying the assembly of mixed GlyRs composed of recombinant mEos4b-GlyRα1 and endogenous GlyRβ subunits. We occasionally also observed mEos4b-GlyRα1 puncta that did not co-localise with gephyrin. These are likely clusters of homopentameric receptors in the extrasynaptic membrane, similar to the clustering of GlyRα3L homopentamers reported in HEK 293 cells (*Notelaers et al., 2012*). In contrast to GlyRα1, expression of GlyRα2 produced a diffuse labelling in most infected neurons with only weak synaptic puncta. The distribution of recombinant GlyRs at synapses and in the extrasynaptic membrane was confirmed by SMLM imaging of the mEos4b-tagged receptor subunits (*Figure 6B*). In line with the above observations, we found a mainly clustered distribution of GlyRα1, both at synapses and in the extrasynaptic membrane. The GlyRα2 subunit had a large extrasynaptic receptor population, likely composed of diffusely distributed GlyRα2 homopentamers, and only a few synaptic heteropentamers. The mEos4b-GlyRβ subunit was visibly expressed only in a few cells, where it had a predominantly synaptic localisation as a result of its direct interaction with gephyrin.

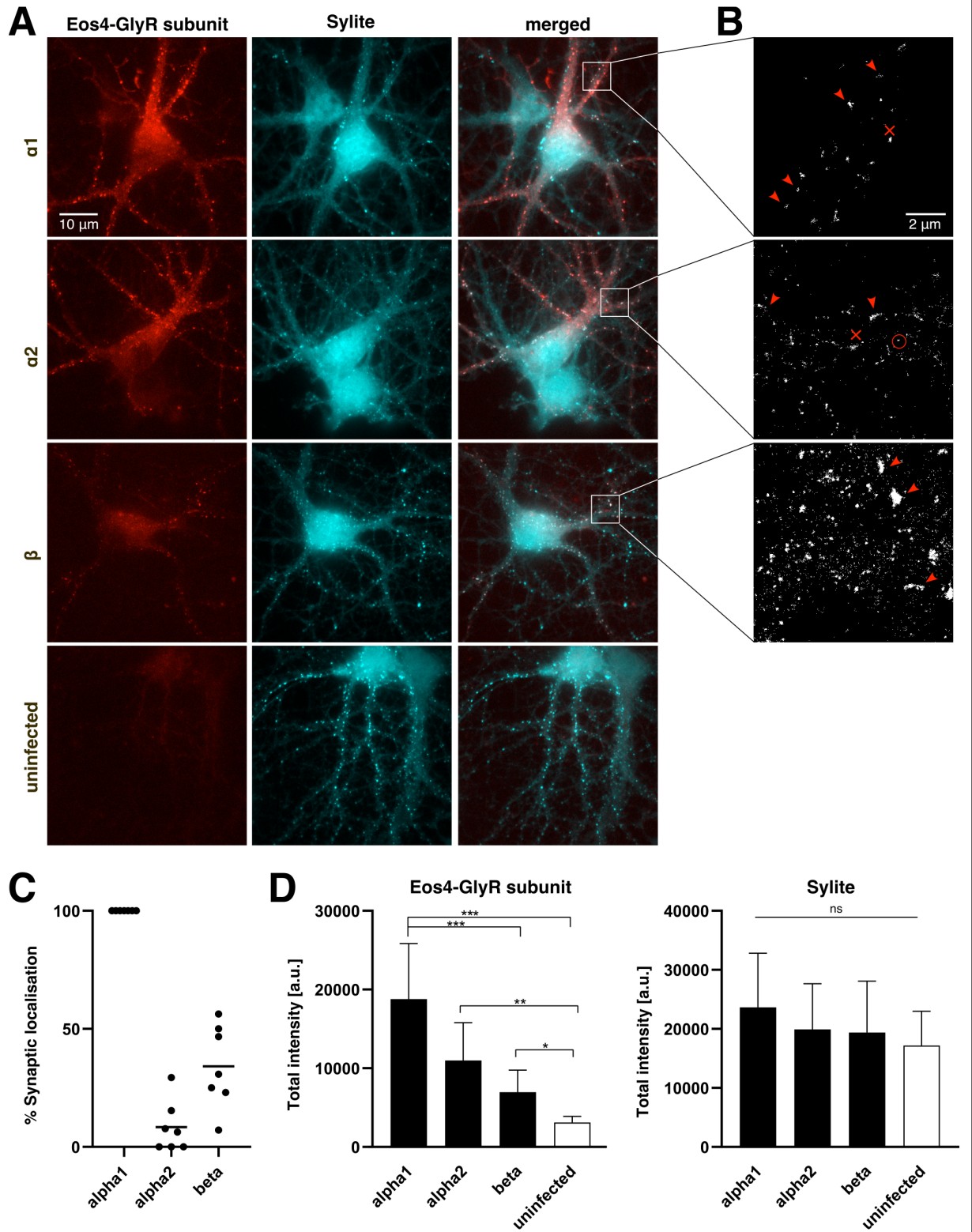

**Figure 6.** Expression of recombinant glycine receptor (GlyR) subunits in cultured hippocampal neurons. (**A**) Cultured mouse embryonic hippocampal neurons (E17.5) were transduced with lentivirus expressing mEos4b-tagged GlyR subunits α1, α2, or β (red), fixed at day in vitro 17 (DIV17) and stained for gephyrin (Sylite marker, cyan). Bottom images: uninfected control neurons. Scale bar: 10 μm. (**B**) Single molecule localisation microscopy (SMLM) pointillist images showing the photoconverted mEos4b detections. Dense clusters of synaptic (red arrowheads) and extrasynaptic receptors (red crosses) are indicated. Diffusely distributed extrasynaptic GlyR complexes (red circle) are seen as small clusters of detections resulting from the repetitive

*Figure 6 continued on next page*

*Figure 6 continued*

detection of a single mEos4b fluorophore. Scale bar: 2 µm. (**C**) Quantification of the percentage of infected neurons displaying mEos4b-positive GlyR clusters that co-localise with synaptic gephyrin clusters. Each data point represents one coverslip of cultured neurons (n=7 coverslips per condition, corresponding to 105 cells for GlyRα1, 114 cells for GlyRα2, and 109 cells for GlyRβ, from N = 3 independent experiments, i.e. cultures). The mean is indicated as a horizontal line. (**D**) Quantification of the total fluorescence intensity of mEos4b-tagged GlyR subunits at Sylite puncta in infected neurons and uninfected controls. The integrated mEos4b fluorescence (left graph) and integrated Sylite fluorescence (right) was measured for every Sylite-positive punctum and the median calculated per cell (n = 21 cells for GlyRα1; 9 for GlyRα2; 33 for GlyRβ, and 18 control cells, from N = 3 experiments; mean ± SD; Kruskal-Wallis [KW] test. *p < 0.05; **p < 0.01; ***p < 0.0001; n.s., not significant). The camera offset was corrected using the minimum pixel intensity in each channel. The signal in the mEos4b channel in the control cultures represents the fluorescence background.

For a more systematic evaluation of the sub-cellular distribution of the GlyRs, we counted the percentage of infected neurons in which a synaptic localisation of the recombinant receptor subunits could be seen in the merged epifluorescence images (*Figure 6C*, see Methods). Essentially all GlyRα1-infected neurons had mEos4b-positive gephyrin clusters, indicating a robust synaptic localisation of the α1 subunit. From this, it can be deduced that most, if not all, hippocampal neurons express endogenous GlyRβ transcripts. When the cultures were infected with mEos4b-GlyRβ, only a few neurons showed a synaptic localisation, meaning that there are only a few endogenous α subunits available and that they constitute the limiting factor for synaptic targeting. Interestingly, very few mEos4b-GlyRα2 expressing neurons displayed synaptic localisation, which suggests that not all α subunits are equally capable of forming heteropentameric GlyR complexes with endogenous β subunits.

We also quantified the total intensity of mEos4b-tagged GlyRα1, α2 and β at synaptic gephyrin clusters in infected hippocampal neurons and uninfected control cultures (*Figure 6D*). Consistent with our previous observations, mEos4b-GlyRα1 accumulated strongly at inhibitory synapses, again pointing to efficient synaptic targeting. GlyRα2 and GlyRβ subunits showed lesser accumulation at synapses. Pairwise comparison between mEos4b-GlyRα1 and mEos4b-GlyRα2 expressing neurons using a non-parametric MW test showed that the mean total intensity of synaptic GlyRα1 is significantly higher than that of GlyRα2 (p < 0.01, MW test). Although the total intensity of the Sylite puncta was not significantly different between conditions when all groups were compared (*Figure 6E*; KW test), a pairwise comparison between mEos4b-GlyRα1 expressing neurons and uninfected controls showed a significant increase in Sylite labelling (p < 0.05, MW test). This seems to indicate that enhanced expression of GlyRα1/β heteropentamers can augment the size of the postsynaptic gephyrin scaffold at inhibitory synapses in the hippocampus.

## Discussion
### Detection of heteropentameric GlyR complexes at hippocampal synapses using SMLM

Making use of the outstanding sensitivity of SMLM, this study describes the presence of very low-copy numbers of GlyRs at synapses throughout the hippocampus. Key to this was the use of highly specific labelling of synaptic receptors using a KI mouse model expressing endogenous mEos4b-GlyRβ that was used either directly for SMLM through the photoconversion of the mEos4b fluorescent protein or as an antigen that was recognised by a specific anti-mEos nanobody. To positively identify the GlyRβ signals, it was necessary to reduce as much as possible the non-specific background of detections that invariably occurs in SMLM recordings. In the case of mEos4b imaging, this was done by acquiring a negative control movie without photoconversion of the fluorophore. In other words, the photophysical properties of the fluorophore can be used to identify them with some certainty (*Wulffele et al., 2022*). Since we focused on GlyRs at inhibitory synapses, the co-localisation of the detections with the synaptic gephyrin scaffold provided additional assurance that single mEos4b fluorophores could be identified in 10-µm-thick slices of brain tissue. These data illustrate the power of SMLM to detect rare or sparsely distributed target molecules in complex samples, in addition to the high spatial precision afforded by super-resolution imaging.

Another strength of SMLM is the possibility to gain access to absolute molecule numbers. Since a fluorophore can be detected several times during the recording, the number of detections has to be translated into the number of emitting fluorophores (*Patrizio and Specht, 2016*). This is generally more straightforward with fluorescent proteins such as mEos4b that are irreversibly bleached after a

number of detections (*Wulffele et al., 2022*), instead of organic fluorophores (e.g. AF647) that can blink over extended periods. In our case, the use of a KI mouse (*Glrb*[eos/eos]) expressing recombinant mEos4b-GlyRβ made it possible to calculate the copy number of endogenous GlyRs at synapses (*Maynard et al., 2021*). To ensure accurate readings, SMLM movies were recorded until all fluorophores were exhausted (typically 5000 frames in hippocampus and as many as 15,000 in the spinal cord, see Methods). Even so, the obtained values have to be taken as estimates given the stochasticity of the number of detections per mEos4b fluorophore.

Our quantification in spinal cord slices indicated that there are on average 120 GlyRs per synapse (*Table 1*). This result is similar to earlier measurements, in which we calculated median values of 114 and 238 GlyRs at synapses in the dorsal and the ventral horn of the spinal cord, respectively (*Maynard et al., 2021*). The slightly lower values compared to the earlier study could be related to the lower age of the animals (40 days instead of 2 or 10 months), or to the greater thickness of the sample (10 μm vs 2 μm slices) that affects the signal-to-noise ratio of the detections. Furthermore, a simpler approach was used in the current study to translate detections into copy numbers. We divided the number of detections at synapses by the number of clusters of detections in sparsely labelled regions in the CA3 that contain mostly individual and spatially separated GlyRs (including extrasynaptic receptors). The advantage of this estimation is that it is independent of the stoichiometry of heteropentameric GlyRs that remains controversial (e.g. *Durisic et al., 2012*; *Durisic et al., 2014*; *Grudzinska et al., 2005*; *Maynard et al., 2021*; *Patrizio et al., 2017*; *Yu et al., 2021*; *Zhu and Gouaux, 2021*). However, our quantification could underestimate the GlyR copy numbers, because a certain fraction of fluorescent proteins like mEos4b is often not functional or not detected under the given imaging conditions, partly due to their complex photo-physical properties (e.g. *Durisic et al., 2014*; *Patrizio et al., 2017*; *Wulffele et al., 2022*).

As opposed to spinal cord synapses, the numbers of GlyRs at hippocampal synapses are very low. In most cases, only a single cluster of detections (i.e. a single heteropentameric GlyR complex) was present at these synapses (*Table 1*). Yet, the existing GlyRs appear to be well integrated into the gephyrin domain as judged by dual-colour super-resolution microscopy, pointing to a possible role at synapses (see below).

## Discrepancies between transcription and protein expression of GlyRs in the brain

Our study addresses a long-standing debate about the presence of GlyRs in the brain. According to many reports, including recent transcriptomic analyses, *Glrb* mRNA is highly expressed in neurons throughout the telencephalon, including the hippocampus (*Ceder et al., 2024*; *Fujita et al., 1991*; *Malosio et al., 1991*; *Figure 1*). For instance, ISH data from mouse brain show strong *Glrb* signals in the CA1 neurons consistent with elevated transcription in pyramidal cells (Allen Brain Atlas, https://mouse.brain-map.org/gene/show/14434). This is also true for the human brain, where high *GLRB* mRNA levels are detected in all sub-regions of the hippocampus (https://www.proteinatlas.org/ENSG00000109738-GLRB/brain; *Karlsson et al., 2021*). In spite of this, the protein expression of GlyRβ receptor subunits in the hippocampus is very low, as judged by our analysis of synaptic (heteropentameric) GlyRs in mEos4b-GlyRβ KI animals. What, then, is the reason for the low protein expression of GlyRβ?

A likely explanation is that the assembly of mature heteropentameric GlyRs depends critically on the co-expression of endogenous GlyR α subunits. The presence of transcripts of the GlyR subunits α1, α2, and α3 in the brain has been demonstrated (*Ceder et al., 2024*; *Malosio et al., 1991*), in line with the existence of extrasynaptic (homopentameric) GlyR complexes (*Chattipakorn and McMahon, 2002*; *Molchanova et al., 2017*; *Mori et al., 2002*; *Song et al., 2006*). Our re-analysis of transcriptomic data confirms that the *Glra1* mRNA levels are low in the telencephalon and increase towards dorsal regions of the brain (*Figure 1A*), in parallel with the expression of GlyRβ protein and its localisation at inhibitory synapses (*Maynard et al., 2021*). This raises the interesting possibility that heteropentameric assembly and subsequent synaptic targeting of GlyRs may depend specifically on the concomitant expression of both *Glra1* and *Glrb* transcripts. To test this hypothesis, we expressed recombinant GlyR α and β subunits in cultured hippocampal neurons. Lentiviral expression of mEos4b-GlyRβ resulted in synaptic receptors only in some cells, suggesting that in the majority of the neurons the *Glrb* transcript is not the limiting element for cell-surface delivery and synaptic

targeting of heteropentameric GlyR complexes. Lentiviral infection with an mEos4b-GlyRα1 construct resulted in efficient surface expression of GlyRα1 clusters in hippocampal neurons, where it generally co-localised with the synaptic gephyrin scaffold. In contrast, mEos4b-GlyRα2 was mostly extrasynaptic and less often and less strongly present at synapses. These observations indicate that GlyRα1 (and possibly GlyRα3) could have a particular role in the assembly and forward trafficking of heteropentameric GlyRs towards the plasma membrane.

Our analyses also showed that lentivirus infection did not alter the gephyrin cluster intensities in hippocampal neurons, suggesting that long-term expression of GlyR subunits does not affect the size of inhibitory synapses per se. However, the expression of mEos4b-GlyRα1 led to a slight increase in total gephyrin intensity. In our view, this is the result of the increased expression of GlyRα1/β heteropentamers and their accumulation at inhibitory synapses, which in turn can recruit additional gephyrin molecules to the postsynaptic scaffold.

## Possible roles of low-copy GlyRs at brain synapses

The low number of GlyRs at hippocampal synapses begs the question of what their role may be. Experiments in *Glra2* knock-out animals have shown that GlyRs help maintain the excitatory/inhibitory balance in the dorsal striatum (*Devoght et al., 2023*). However, these receptors are likely to be extrasynaptic GlyRα2 homopentamers. On the other hand, the contribution of a single synaptic GlyR to the chloride influx during inhibitory neurotransmission is probably not significant. In line with this interpretation, glycinergic miniature IPSCs in hippocampal slices are not generally detected (*Chattipakorn and McMahon, 2002*; *Mori et al., 2002*; *Song et al., 2006*; but see *Muller et al., 2013*). The same is true for the *putamen* (dorsal striatum), where whole-cell currents were recorded in response to glycine application (*Molchanova et al., 2017*). Again, our quantification indicates that GlyR copy numbers in this area are low (*Table 1*). Synaptic GlyRs are more numerous at synapses in the *nucleus accumbens* (ventral striatum), and indeed, our electrophysiological data support this finding in agreement with earlier studies (*Muñoz et al., 2018*).

Another possible role of low-copy GlyRs at synapses in the hippocampus, dorsal striatum, and maybe other regions of the telencephalon could be a structural one. This concept is based on the high affinity of the GlyRβ-gephyrin interaction (*Kasaragod and Schindelin, 2018*) that stabilises the receptor, as well as the gephyrin scaffold at inhibitory synapses (*Chapdelaine et al., 2021*). Accordingly, overexpression of synaptic GlyRα1/β heteropentamers led to an increase in gephyrin levels at hippocampal synapses (*Figure 6D*). Our two-colour SMLM data of endogenous mEos4b-GlyRβ and gephyrin at hippocampal synapses further confirm that despite their low number, the GlyRs are integral components of the postsynaptic gephyrin domain in support of a structural role. GABA$_A$R subunits have a much lower affinity for gephyrin and cannot provide the same level of stability (e.g. *Kostrz et al., 2024*; *Maric et al., 2014*; *Maric et al., 2011*). However, GABA$_A$Rs could probably be recruited efficiently to an existing gephyrin scaffold.

In conclusion, our data positively identify the presence of very small numbers of heteropentameric GlyRs at inhibitory synapses in the brain. This is drastically different to the situation in the *nucleus accumbens*, and even more so in the spinal cord, where GlyRs are abundant and densely clustered at most inhibitory synapses. While it is reasonable to classify different inhibitory synapses as mainly glycinergic or GABAergic, it should be noted that this is a simplification that does not account for the full diversity of inhibitory synapses that may assemble in a continuum of mixed compositions across the entire dynamic range.

# Methods

## Single-cell transcriptomic analysis

### Data availability

The Allen Mouse Brain transcriptomic data utilised in this study are available through the Gene Expression Omnibus (GEO) database under accession code GSE246717. Specifically, 10× single-cell RNA sequencing data of four datasets, each representing a distinct brain region, ventral striatum, dorsal striatum, hippocampus, and medulla, were accessed through the Sequence Read Archive (SRA) under their run accession codes SRR26528931, SRR26528889, SRR26528942, and SRR26528896, respectively, and retrieved in FASTQ format using the SRA Toolkit (v2.11.0).

## Data pre-processing and quality control

Sequencing data were aligned to the mouse genome (mm10, 10x Genomics version 2020-A) using Cell-Ranger (v7.1.0) with default parameters. The filtered gene expression count matrices were then analysed in R (v4.4.1) using the Seurat package (v5.0.1). Each dataset underwent individual quality control (QC) to remove low-quality data and doublets. Briefly, cells were filtered out if they contained <1000 expressed genes or >10% of transcripts derived from mitochondrial genes. Any gene that appeared in <10 cells was removed. Doublets were removed using the DoubletFinder package. From around 35,000 cells, approximately 26,000 cells passed QC.

## Integration and cell-type annotation

The merged Seurat object was normalised using the SCTransform function, with the vst method set to v1 and the rest of parameters set to default, followed by linear dimensionality reduction with PCA. To determine how many principal components to use in downstream analysis, two criteria were used: (1) the cumulative percent variation explained was >90% and the individual percent variation explained was <5%; (2) the change in percent variation was >0.1%. Data integration was performed using the RPCA method, which was selected to account for batch effects across datasets from different brain regions. Further dimensionality reduction and clustering were performed in accordance with the standard Seurat workflow and with the following parameters: UMAP visualisation was performed on the integrated data using the 30 most significant PCs. FindNeighbors was applied on the 2D UMAP embedding (using 'umap.rpca' reduction) instead of PCA (default), followed by clustering with FindClusters at a resolution of 0.05, selected to yield a small number of broad clusters corresponding to major mouse brain cell types. Cell-type identities were assigned to the resulting clusters based on the expression of canonical marker genes for neurons (*Snap25, Rbfox3, Syp, Snhg11*), oligodendrocytes (*Mobp, Mog, Plp1, Mag, Mbp*), oligodendrocyte precursor cells (*Vcan, Pdgfra, Cspg4, Gpr17*), astrocytes (*Gfap, Aldh1l1, Fgfr3, Col23a, Aqp4, Slc1a2, Trp63, Slc7a10, Atp1a2, Gja1*), and microglia (*Ctss, Csf1r, Ptprc, Itgam, Ly86, Myo1f*) drawn from the literature.

## Primary culture of hippocampal neurons

Hippocampal neurons were prepared from wild-type mice with Swiss background, at embryonic day E17.5. Experiments were performed in accordance with the European Directive on the protection of animals used for scientific purposes (2010/63/EU) and the regulations of the local veterinary authority (Inserm UMS44-Bicêtre, licence G94043013). The mice came from excess production for another project, meaning that no animals were generated for the current project.

Pregnant mice were put to death by cervical dislocation, and the embryos collected by caesarean section and decapitated. Hippocampi were rapidly dissected in cold Hank's Balanced Salt Solution (HBSS, Gibco, #14180-046) containing 20 mM HEPES (Gibco, #15630-056) and incubated at 37°C for 15 min in dissection medium containing 0.25% trypsin (Gibco #15090-046). After trypsinisation, hippocampi were washed twice in plating medium composed of Minimal Essential Medium (MEM) containing Earle's Balanced Salts (EBSS) (Cytiva, #SH30244.01), 2 mM GlutaMAX (Gibco, #35050-038), 1 mM sodium pyruvate (Thermo Fisher Scientific, #11360-039), and 10% heat-inactivated horse serum (Gibco, #26050-088). The hippocampal tissue was then triturated in plating medium containing 0.3 mg/ml DNase I (Merck, #11284932001). Neurons were seeded at a concentration of $3.4 \times 10^5$/cm$^2$ in 12-well plates (Thermo Fisher, #150628) on round-glass coverslips (type 1.5, 18 mm diameter; Marienfeld, #0112580) that were pre-coated with poly-D,L-ornithine (Merck, #P8638). The medium was replaced 4 hr after plating with maintenance medium: Neurobasal medium (Gibco, #21103-049) containing B-27 supplement (Gibco, #17504-044) and 2 mM GlutaMAX. Once a week, 300 µl of fresh maintenance medium was added. Hippocampal neurons were typically infected with ≤50 µl of lentivirus stocks at DIV3 (FU-mEos4b-GlyRα2, FU-mEos4b-GlyRβ) or DIV10 (FU-mEos4b-GlyRα1) and fixed for immunocytochemistry at DIV16.

## Lentivirus expression constructs

The following lentivirus constructs were used for the expression of mEos4b-tagged GlyR subunits: FU-mEos4b-GlyRα1, expressing the coding sequence (cds) of rat *Glra1* isoform a (UniProt P07727-1) (*Patrizio et al., 2017*) and FU-mEos4b-GlyRβ-bis the cds of human *GLRB* (UniProt P48167-1) (*Grünewald et al., 2018*). The receptor sequence in construct FU-mEos4b-GlyRβ-bis (excluding the

signal peptide and mEos4b sequence) was replaced with the cds of human *GLRA2* (UniProt P23416) to generate the expression construct FU-mEos4b-GlyRα2.

For virus production, HEK-293 tsA201 cells were co-transfected with equal amounts (5 µg each) of the replicon plasmid and the three helper plasmids pMDLg/pRRE, pRSV-Rev, and pMD2.G (Addgene #12251, #12253, #12259) using Lipofectamine 2000 (Invitrogen, #11668-019). Cells were cultured in maintenance medium (see above) supplemented with 5 U/ml penicillin and 5 µg/ml streptomycin at 37°C/5% $CO_2$ for 24 hr, at which point the medium was exchanged. The culture medium containing lentivirus was collected at 48–55 hr, filtered with a pore size of 0.45 µm, and frozen as aliquots at –70°C.

## Immunocytochemistry in cultured neurons
Hippocampal neurons were fixed at DIV16 with 4% wt/vol PFA and 1% wt/vol sucrose in 0.1 M phosphate buffer (PB), pH 7.4 for 10 min. After three washes in PBS, the cells were permeabilised in PBS, pH 7.4 containing 0.25% Triton X-100 and 4% wt/vol bovine serum albumin (BSA) (Sigma, #A7030) for 10 min and then blocked in PBS containing 4% wt/vol BSA for 1 hr. Neurons were incubated for 1 hr with the 200 nM of the gephyrin marker Sylite (*Khayenko et al., 2022*) in PBS containing 1% wt/vol BSA. The cells were rinsed twice in PBS and kept in PBS overnight at 4°C until imaging.

## Sample preparation of spinal cord and brain slices for SMLM of endogenous mEos4b-GlyRβ
Brain and spinal cord tissue from homozygous mEos4b-GlyRβ KI mice (*Glrb*eos/eos, mouse strain C57BL/6N-*Glrb*tm1lcs, backcrossed into C57BL/6J, accession number MGI:6331106; *Maynard et al., 2021*) was recovered from an earlier project (*Wiessler et al., 2024*). In this mouse strain, the β subunit of the GlyR is tagged at its N-terminus with the photoconvertible mEos4b fluorescent protein. The chosen animals did not carry any genomic modification other than *Glrb*eos/eos. For control experiments, brain tissue from a wildtype Swiss mouse was used. Freshly frozen spinal cords and brains from male and female mice at postnatal day 40 were embedded in OCT (Pink Neg-50, Thermo Fisher Scientific, #6502P) and coronal slices of a nominal thickness of 10 µm were cut on a cryostat (Leica, Wetzlar, Germany, #CM3050S) with a chamber temperature of –23°C. The slices were collected on SuperFrost Plus glass slides (Epredia, #J7800AMNZ), fixed with 2% wt/vol PFA and 0.5% wt/vol sucrose in PB, pH 7.4 for 10 min, and rinsed three times for 5 min in PBS at room temperature (RT). Afterwards, the slices were permeabilised and blocked in PBS containing 4% wt/vol BSA (A7030, Sigma) and 0.25% vol/vol Triton X-100 for 15 min. Optionally, the slices were labelled overnight with a polyclonal antibody against NeuN (raised in chicken, Synaptic Systems, #266006, RRID:AB_2571734, 1:2000) at 4°C, followed by 4 hr of AF488-conjugated donkey anti-chicken secondary antibody (Jackson ImmunoResearch, #703-545-155, RRID:AB_2340375, 1:2000) at RT. The gephyrin-specific peptide probe Sylite (*Khayenko et al., 2022*) was applied at a final concentration of 50 nM in PBS containing 1% wt/vol BSA for 1 hr at RT. After two washes of 5 min in PBS, slices were mounted in PBS, covered with a glass coverslip (type 1.5), sealed with PicoDent Twinsil Speed (#1300-1002) and kept overnight at 4°C or directly used for SMLM imaging of the mEos4b photoconvertible protein.

## SMLM of mEos4b fluorescent protein
SMLM imaging of mEos4b-tagged GlyRs was performed using an ELYRA PS.1 microscope setup (Zeiss, Jena, Germany). Samples were placed on the motorised stage of an Axio Observer.Z1 SR inverted microscope and imaged with a Plan-Apochromat 63×/NA 1.4 oil immersion objective, with an additional 1.6× lens in the emission path. Images were captured with an Andor iXon 897 back-thinned EMCCD camera (16 bit, 512×512 pixels, 16 µm pixel size, QE 90%, set at –60°C working temperature), resulting in an image pixel size of 160 nm. Reference images in the green channel were taken with a 488 nm excitation laser (nominal output 300 mW) and a band pass (BP) 495–575 nm (+LP 750) emission filter. For the far-red channel, we used a 642 nm excitation laser (nominal output 150 mW) and a long pass (LP) 655 nm emission filter. SMLM recordings were performed by exploiting the properties of the fluorescent protein mEos4 that is photoconverted from green to red state upon UV illumination (405 nm laser, nominal power 50 mW) and image acquisition in the red channel (561 nm laser, nominal power 200 mW, emission filter BP 570–650+LP 750). The localisation precision of each detection was

calculated from the fitting parameters and was obtained directly from the Zeiss NEO software. For the experiments reported here, the localisation precision was 33.1±12.3 nm in x/y (mean ± SD).

Regarding the experiments in hippocampal cultures (data in *Figure 6*), infected neurons expressing mEos4b-tagged GlyR subunits were identified in the green channel, and single reference epifluorescence images of 100 ms exposure were taken with the 488 nm excitation laser set at 1% and the 405 nm laser at 1% of the maximal power, and a camera gain of 100. For Sylite, we used the far-red channel, taking one image of 100 ms with the 642 nm laser at 10% output and a camera gain of 10. An SMLM movie of 10,000 frames was then recorded at 20 Hz streamed acquisition (50 ms frames) with constant 561 nm laser illumination at 100% laser output corresponding to a maximal power density of 0.95 kW/cm$^2$ and a gain of 300. The photoconversion of mEos4b-tagged GlyR subunits (from green to red) was done by continuous 405 nm laser illumination that was gradually increased from 0.01% to 4% intensity ($\leq$5.3 W/cm$^2$ irradiance).

For the detection of endogenous mEos4b-GlyRβ in brain slices (data in *Figures 2 and 4*), we used the following acquisition parameters: hippocampal and striatal regions were identified using NeuN immunolabelling in the green channel (AF488). A single epifluorescence image of 50 ms was taken with 488 nm illumination, 0.2% laser intensity, and a camera gain of 200 to record the NeuN immunofluorescence. Sylite was detected in the far-red channel, taking one image of 100 ms at 642 nm (5% output, camera gain 100). A first SMLM recording of 2000 frames at 20 Hz was done in the red channel with 561 nm laser illumination at 80% (irradiance 0.8 kW/cm$^2$), a camera gain of 300, and without 405 nm laser illumination. Under these conditions, mEos4b is not converted into the active (red) form. Afterwards, a second SMLM movie of 5000 frames was taken with the same settings, but with the addition of continuous 405 nm laser illumination that was gradually increased from 0.01% to 3% intensity (irradiance $\leq$4 W/cm$^2$). During these recordings, the total population of mEos4b molecules is converted (and bleached), since only very sparse detections are seen at the end of the movie. Glycinergic synapses in spinal cord slices (*Figure 2—figure supplement 1*) were identified as bright puncta of (unconverted) mEos4b-GlyRβ subunits in the green channel, and reference images were taken (green, mEos4b: 100 ms acquisition with the 488 laser at 0.2% power, camera gain 200; far-red, Sylite: 100 ms, 642 laser at 3%, gain 200 ms). SMLM movies of 15,000 frames were recorded with the same settings as before with a continuous 405 nm laser illumination, gradually increased from 0.01% to 5% intensity to ensure the complete conversion of mEos4b by the end of the experiment.

## Epifluorescence image analysis in cultured neurons

For the analysis of cultured hippocampal neurons, only transduced cells expressing mEos4b-GlyRα1, α2, or β were considered (data in *Figure 6*). Where the expression of the recombinant GlyR subunit was not obvious, this was confirmed by SMLM (*Figure 6B*, see below). Within this population, the fraction of cells exhibiting synaptic localisation of GlyR was determined as the number of cells in which mEos4b signals co-localised with Sylite-positive gephyrin clusters, as judged from the epifluorescence images (*Figure 6A*). The fluorescence intensity of mEos4b-GlyRα1, α2, and β puncta at synapses was measured only in the infected neurons, showing a synaptic localisation of the subunits. Synaptic puncta labelled with Sylite were detected using the spot detector plugin (Icy; *de Chaumont et al., 2012*) with the following parameters: bright spots over dark background, detection scale 2 (sensitivity 40), and a size filter between 4 and 100 pixels. To restrict the analyses to synaptic puncta from infected neurons, the signal intensity in the mEos4b channel was thresholded using the mean+2 SD calculated from uninfected control neurons. The pixel intensity was also corrected for the camera offset, by subtracting the minimum pixel intensity (1500 a.u.) from all images.

## SMLM image analysis of mEos4b fluorescent protein

SMLM movies were processed with Zen software (Zeiss, Zen 2012 SP5 FP3 black, 64 bit) using a peak mask size of 7 pixels, a peak intensity of 6, and excluding overlapping molecules. The single molecule localisations were corrected for x/y drift with model-based algorithm (without fiducial markers). A rendered super-resolution image was reconstructed from the pointillist image, in which each detection is represented with a two-dimensional Gaussian distribution with a width corresponding to its point spread function and a 10 nm pixel size and saved in TIFF format.

Inhibitory synapses in spinal cord and brain slices were identified using the gephyrin-specific peptide marker Sylite (*Khayenko et al., 2022*). First, the Sylite reference images were adjusted to a

10 nm pixel size by multiplying the pixel number by a factor of 16. We then detected Sylite puncta using the spot detector plugin in Icy with these parameters: bright spots over dark background were detected at scale 5 set to a sensitivity of 100, and the size filtered between 400 and 10,000 pixels. The obtained regions of interest (ROIs) of Sylite were then employed as masks to measure the intensity of the mEos4b-GlyRβ signals in the rendered SMLM image. Where necessary, the alignment of the reference image and the rendered SMLM image was manually adjusted in ImageJ. To test the specificity of the co-localisation between Sylite and mEos4b-GlyRβ, we performed a pixel shift analysis with images of the CA3 region in $Glrb^{eos/eos}$ slices by horizontally flipping the Sylite channel relative to the mEos4b channel. The number of mEos4b-GlyRβ detections per gephyrin cluster was then recalculated using Icy software. The output data consist of an Excel table containing the total intensity of Sylite and mEos4b-GlyRβ in each ROI. To convert the mEos4b-GlyRβ signals into detection numbers, the obtained total intensity values per ROI were divided by the integrated intensity of single mEos4b detections in the image.

A similar approach was used to estimate the copy numbers of mEos4b-GlyRβ containing GlyRs at inhibitory synapses in the CA3 region of the hippocampus. Rendered SMLM images of mEos4b-GlyRβ with a pixel size of 10 nm were segmented in Icy (detection of bright spots over dark background, scales 3 and 4 at a sensitivity of 100, and filtering between a size of 400 and 10,000 pixels). Due to the low mEos4b-GlyRβ density in CA3, the obtained ROIs were considered to be clusters of detections arising from mEos4b fluorophores of a single heteropentameric GlyR (independent of its subunit stoichiometry). The average number of detections per cluster was then calculated by dividing the total intensity of the cluster by the total intensity of a single detection. This value was used as a conversion factor to translate detection numbers into molecule numbers and was applied to all SMLM images from spinal cord, hippocampus, and striatum. The copy numbers were further corrected by background subtraction using the negative control (CA3 region in wildtype C57BL/6J slices) and are given in *Table 1*.

## Immunostaining of brain tissue slices used dual-colour SMLM with organic fluorophores

Frozen brain tissue from $Glrb^{eos/eos}$ mice (*Maynard et al., 2021*; *Wiessler et al., 2024*) was cut into 10-μm-thick coronal cryostat sections and processed as described above. For dual-colour SMLM with organic fluorophores, the slices were labelled with primary antibodies against gephyrin (mAb7a mouse monoclonal, Synaptic Systems, #147011, RRID:AB_2810215, 1:1000 dilution) and mEos protein (AF647-conjugated FluoTag-X2, NanoTag Biotechnologies; #N3102-AF647-L, RRID:AB_3076063, 1:500) overnight at 4°C, followed by secondary goat anti-mouse antibody coupled with a single CF680 dye (Biotium, #20817, 1:1000) for 4 hr at RT. After washing in PBS, the slices were kept overnight at 4°C and imaged the next day.

## Dual-colour SMLM acquisition and image processing

The cover glasses were placed in a closed Ludin chamber (Life Imaging Services) and, through the perfusion holes, was added a homemade dSTORM buffer (*Yang and Specht, 2020*) that was prepared as follows: A suspension containing 200 μg catalase (from bovine liver, Merck, # C30 1003493507) was washed three times with 1 ml of cold PBS and collected by centrifugation at 12,000×$g$ at 4°C for 1 min. After removing the supernatant, the catalase crystals were resuspended in 1 ml of PBS and incubated at 37°C with agitation for 30 min. The final dSTORM buffer was composed of PBS pH 7.4, containing 50 mM cysteamine hydrochloride (=β-mercaptoethylamine, MEA, Merck, #M6500), 250 mM glucose (Merck, #G7021), 0.5 mg/ml glucose oxidase (from *Aspergillus niger*, Merck, #G2133-10KU), and 40 μg/ml dissolved catalase. Prior to use, the dSTORM buffer was degassed with $N_2$, transferred into a syringe, and kept on ice.

SMLM experiments were carried out with an Abbelight nanoscope (SAFe 360 Nexus SD) installed on the Zeiss Elyra PS1 setup described above, using two sCMOS cameras (Hamamatsu Orca-Fusion BT) for simultaneous dual-colour imaging. Before the acquisitions, the two cameras were aligned using 0.1 μm TetraSpeck beads (Invitrogen, #T-7279) deposited on a glass coverslip. The Ludin chamber with the tissue slices in dSTORM buffer was then placed on the stage of the inverted microscope and imaged with a Plan-Apochromat 100×/NA 1.46 oil-immersion objective without additional magnification. The far-red AF647 and CF680 dyes were excited with a 640 nm laser of 520 mW nominal power

(Oxxius LPX-640-500) using adaptable scanning of the excitation region (ASTER). SMLM images were acquired using NEOimaging software v.2.17.1 with the following parameters: 10,000 images (image size 256×256 pixels and 97 nm pixel size) of 50 ms exposure were taken in the far-red channel with 80% laser power and a field of excitation set at 10%, resulting in a maximal irradiance of 20.3 kW/cm². Fluorophore blinking was gradually adjusted with a 405 nm laser (LBX-405-100, nominal output 108 mW), increasing the power from zero to 2% intensity along the recording (≤45 W/cm² irradiance). The emitted wavelengths were separated into two light paths with a 700 nm dichroic mirror for simultaneous dual-colour imaging and filtered with a BP from 669 nm to 741 nm.

STORM images were processed with NEO Analysis software (Abbelight v.39). The raw TIFF movies from the transmitted camera (>700 nm) and the reflected camera (<700 nm) were loaded and processed sequentially. Single fluorophore signals were detected using temporal mean subtraction with a sliding window of 50 frames, and fit with a Gaussian distribution. The detection coordinates in both channels were corrected for x/y drift and saved as coordinate tables. For spectral demixing and image reconstruction (*Figure 3—figure supplement 2*), the detections from the two cameras were imported in the NEO 3D viewer, superimposed and automatically re-aligned. The intensity ratios for each detection were calculated according to the formula $I_{reflected}/(I_{reflected} + I_{transmitted})$, and we selected for each dye (AF647 and CF680) the inferior and superior cut-offs (0.25–0.32 for AF647; 0.42–0.70 for CF680). After removing the detections that had a precision ≥25 nm, the spectrally demixed data were saved as coordinate tables and in the form of rendered super-resolution images with a normalised Gaussian representation of each detection and a pixel size of 10 nm. The mean localisation precision was calculated in NEO (Abbelight v.39) and was 10.9 ± 4.4 (mean ± SD) for AF647 (anti-mEos nanobody) and 12.2 ± 4.9 for CF680 (anti-gephyrin).

The sub-synaptic distribution of GlyRs (AF647-labelled mEos4b-GlyRβ) and gephyrin (mAb7a-CF680) in hippocampal synapses in the CA3 region was investigated by DBSCAN cluster analysis implemented in NEO 3D viewer (*Khayenko et al., 2022*) using the following parameters: a radius of ε = 200 nm and n ≥ 5 neighbours for the AF647 detections (GlyRβ) and ε = 80 nm and n ≥ 50 neighbours for CF680 (gephyrin). We exported the data table containing the filtered clusters and their coordinates and calculated the Euclidean distance between the CM of the corresponding GlyR and gephyrin clusters, as well as the ratio of the GlyRβ-gephyrin distances divided by the RG of the gephyrin cluster. Values <1 indicate that the GlyRβ detections are closer to the CM of gephyrin than the dispersion (RG) of the gephyrin detections themselves, suggesting that the GlyRs are integrated within the postsynaptic gephyrin cluster.

## Confocal microscopy

For the control experiments shown in *Figure 3—figure supplement 1*, spinal cord slices from *Glrb*^eos/eos and wildtype animals (*Glrb*^WT/WT) were labelled with AF647-conjugated nanobody (1:1000, NanoTag) and with an antibody against gephyrin (mouse anti-gephyrin mAb7a, Synaptic Systems, 1:1000), followed by a donkey anti-mouse secondary antibody coupled with CF568 (Sigma #SAB4600075, 1:1000). Confocal images were taken on an SP8 microscope (Leica) using a 63× oil immersion objective and a Hybrid detector (HyD3). Acquisition of images (512×512 px, 16 bit) with a pixel size of 120 nm in the x/y plane and 1.27 μm in z (pinhole 1) was done in the three channels, green (mEos4b), red (CF568), and far-red (AF647).

## Statistical analyses

Statistical analyses and graphing were performed using GraphPad Prism v.9. Data are represented as mean ± SD (standard deviation) or mean ± SEM (standard error of the mean) as indicated. Statistical significance was calculated using a non-parametric one-tailed or two-tailed MW test, or a non-parametric KW test (one-way ANOVA) with a post hoc Dunn's multiple comparison test.

## Preparation of acute brain slices

Adult C57BL/6J mice (postnatal days 35–41) were anaesthetized with isoflurane and decapitated. The brains were quickly removed on ice and glued to the cooled stage of a vibratome (Leica VT 1200S). Coronal brain sections containing both dorsal and ventral striatum (150 μm) were sliced in ice-cold cutting solution (140 mM choline chloride, 26 mM $NaHCO_3$, 10 mM glucose, 7 mM $MgCl_2$, 2.5 mM KCl, 1.25 mM $NaH_2PO_4$, 0.5 mM $CaCl_2$, saturated with 95% $O_2$ and 5% $CO_2$). Brain slices were placed

in recovery solution (120 mM NaCl, 2.5 mM KCl, 2 mM CaCl$_2$, 2 mM MgCl$_2$, 10 mM glucose, 26 mM NaHCO$_3$, 1.2 mM NaH$_2$PO$_4$, saturated with 95% O$_2$ and 5% CO$_2$) for 45 min at 32°C. Slices were kept at 32°C and used 1–4 hr after slicing.

## Electrophysiology

Acute brain slices were placed in the recording chamber and perfused with oxygenated aCSF (124 mM NaCl, 4.5 mM KCl, 2 mM CaCl$_2$, 10 mM glucose, 1 mM MgCl$_2$, 26 mM NaHCO$_3$, 1.2 mM NaH$_2$PO$_4$, saturated with 95% O$_2$ and 5% CO$_2$, pH 7.4, at RT). MSNs in the dorsal or ventral striatum were visualised using 4× air and 40× water immersion objectives of an upright Zeiss microscope (Axioskop 2FS Plus). A P-1000 micropipette puller (Sutter Instruments) was used to prepare filament-containing borosilicate glass patch pipettes (Hilgenberg GmbH) with a resistance of 4–5 MΩ. Patch pipettes were filled with internal solution (120 mM KCl, 4 mM MgCl$_2$, 10 mM HEPES, 1 mM EGTA, 5 mM lidocaine N-ethyl bromide, 0.5 mM Na$_2$GTP, 2 mM Na$_2$ATP, adjusted to 280 mOsm, pH 7.4). Cells were recorded using voltage clamp at a holding potential of –60 mV in whole-cell configuration with a sampling rate of 10 kHz. During recordings, aCSF bath perfusion was applied for 1 min. To pharmacologically isolate glycinergic mIPSCs, blockers of AMPA receptors (DNQX, 10 µM), nicotinic receptors (DHBE, 0.1 µM), NMDA receptors (L-689560, 5 µM), action potential firing (tetrodotoxin, 0.5 µM), and GABA$_A$Rs (bicuculline, 10 µM) were added to aCSF and recorded for 5 min. To confirm that these mIPSCs were indeed glycinergic currents, strychnine was added at a concentration of 1 µM to the cocktail of blockers and recorded for an additional 1 min. All recordings were acquired using a Multiclamp 700B amplifier (Axon Instruments), stored using 1440A Digidata (Axon Instruments), and analysed using Clampfit 10.7.0.3 (Axon Instruments) and NeuroExpress 24.c.16 (*Szücs, 2022*).

## Acknowledgements

We thank Hans Maric (Virchow Center, University of Würzburg) for the Sylite probe, Carmen Villmann and Natascha Schäfer (Universitätsklinikum Würzburg) for the collection of brain and spinal cord tissue, Krishna Gaete Riquelme (Universidad de Concepción, Chile) for technical assistance, Nicolas Froger (MAPREG, France) for the supply of mouse embryos, Elisabeth Piccart and Petra Bex (UHasselt, Belgium) for brain slice patch clamp training, and Fabrice Schmitt (Zeiss) and Michael Schumacher (NeuroBicêtre, Inserm, UPSaclay) for their generous help with the setting up of the super-resolution imaging platform. We also acknowledge the proficient technical support by Zeiss and Abbelight. This research was funded by the Agence Nationale de la Recherche (ANR-20-CE11-0002, InVivoNanoSpin, to CGS), Inserm (IRP RESiSTER, to CGS), and Fonds voor Wetenschappelijk Onderzoek (11N1422N, V419721N, to BB). SC was supported by a postdoctoral contract through the ANR projects InVivoNanoSpin (ANR-20-CE11-0002) and NeurATRIS (ANR-11-INBS-0011). The SMLM setup was acquired with the support of the Fédération pour la Recherche sur le Cerveau (FRC/Neurodon, opération Rotary – Espoir en Tête, to CGS), and an ERM equipment grant from UPSaclay (to CGS). We also acknowledge support of the international exchange programme ECOS Sud (C21S02, to CGS and Gonzalo Yévenes, UdeC, Chile).

## Additional information

### Funding

| Funder | Grant reference number | Author |
| --- | --- | --- |
| Agence Nationale de la Recherche | ANR-20-CE11-0002 | Christian G Specht |
| Fonds Wetenschappelijk Onderzoek | 11N1422N | Bert Brône |
| Fonds Wetenschappelijk Onderzoek | V419721N | Bert Brône |

| Funder | Grant reference number | Author |
|---|---|---|
| Institut National de la Santé et de la Recherche Médicale | IRP RESiSTER | Christian G Specht |
| NeurATRIS | ANR-11-INBS-0011 | Serena Camuso |
| FRC/Neurodon, opération Rotary – Espoir en Tête 2022 | | Christian G Specht |
| UPSaclay | ERM 2022 | Christian G Specht |
| International Exchange Programme ECOS Sud | C21S02 | Christian G Specht |

The funders had no role in study design, data collection and interpretation, or the decision to submit the work for publication.

## Author contributions

Serena Camuso, Conceptualization, Formal analysis, Investigation, Writing – original draft, Writing – review and editing; Yana Vella, Formal analysis, Investigation, Writing – review and editing; Souad Youjil Abadi, Formal analysis, Writing – review and editing; Clémence Mille, Methodology; Bert Brône, Conceptualization, Funding acquisition, Methodology, Project administration, Writing – review and editing; Christian G Specht, Conceptualization, Funding acquisition, Methodology, Writing – original draft, Project administration, Writing – review and editing

## Author ORCIDs

Serena Camuso https://orcid.org/0000-0002-3116-6340
Yana Vella https://orcid.org/0000-0002-8592-1946
Souad Youjil Abadi https://orcid.org/0009-0007-4645-8008
Clémence Mille https://orcid.org/0000-0003-2069-8250
Bert Brône https://orcid.org/0000-0002-4851-9480
Christian G Specht https://orcid.org/0000-0001-6038-7735

Reviewer #1 (Public review): https://doi.org/10.7554/eLife.109447.2.sa1
Reviewer #2 (Public review): https://doi.org/10.7554/eLife.109447.2.sa2
Reviewer #3 (Public review): https://doi.org/10.7554/eLife.109447.2.sa3
Author response https://doi.org/10.7554/eLife.109447.2.sa4

# Additional files

## Supplementary files
MDAR checklist

## Data availability

The experimental data have been deposited in Figshare (https://doi.org/10.6084/m9.figshare.30912920). The materials generated for this study are available from the corresponding author upon reasonable request.

The following dataset was generated:

| Author(s) | Year | Dataset title | Dataset URL | Database and Identifier |
|---|---|---|---|---|
| Camuso S, Specht C, Brône B, Vella Y | 2026 | Dataset for the article: Single molecule counting detects low-copy glycine receptors in hippocampal and striatal synapses | https://doi.org/10.6084/m9.figshare.30912920 | figshare, 10.6084/m9.figshare.30912920 |

The following previously published dataset was used:

| Author(s) | Year | Dataset title | Dataset URL | Database and Identifier |
|---|---|---|---|---|
| Yao et al | 2026 | A high-resolution transcriptomic and spatial atlas of cell types in the whole mouse brain | https://www.ncbi.nlm.nih.gov/geo/query/acc.cgi?acc=GSE246717 | NCBI Gene Expression Omnibus, GSE246717 |

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
