## [Editor Report · eLife Assessment]

The study presents **convincing** quantitative evidence, supported by appropriate negative controls, for the presence of low-abundance glycine receptors (GlyRs) within inhibitory synapses in telencephalic regions of the mouse brain. Using sensitive single-molecule localization microscopy of endogenously tagged GlyRs, the authors reveal previously undetected populations of these receptors. Although the functional significance of these low-abundance GlyRs remains to be established, the findings offer **valuable** insights and methodologies that will be of interest to neuroscientists studying inhibitory synapse biology.

[Editors' note: this paper was reviewed by Review Commons.]

---

## [Referee Report · Reviewer #1 (Public review)]

Summary:

In this manuscript, the authors investigate the nanoscopic distribution of glycine receptor subunits in the hippocampus, dorsal striatum, and ventral striatum of the mouse brain using single-molecule localization microscopy (SMLM). They demonstrate that only a small number of glycine receptors are localized at hippocampal inhibitory synapses. Using dual-color SMLM, they further show that clusters of glycine receptors are predominantly localized within gephyrin-positive synapses. A comparison between the dorsal and ventral striatum reveals that the ventral striatum contains approximately eight times more glycine receptors and this finding is consistent with electrophysiological data on postsynaptic inhibitory currents. Finally, using cultured hippocampal neurons, they examine the differential synaptic localization of glycine receptor subunits (α1, α2, and β). This study is significant as it provides insights into the nanoscopic localization patterns of glycine receptors in brain regions where this protein is expressed at low levels. Additionally, the study demonstrates the different localization patterns of GlyR in distinct striatal regions and its physiological relevance using SMLM and electrophysiological experiments. However, several concerns should be addressed.

Specific comments on the original version:

(1) Colocalization analysis in Figure 1A. The colocalization between Sylite and mEos-GlyRβ appears to be quite low. It is essential to assess whether the observed colocalization is not due to random overlap. The authors should consider quantifying colocalization using statistical methods, such as a pixel shift analysis, to determine whether colocalization frequencies remain similar after artificially displacing one of the channels.

(2) Inconsistency between Figure 3A and 3B. While Figure 3B indicates an ~8-fold difference in the number of mEos4b-GlyRβ detections per synapse between the dorsal and ventral striatum, Figure 3A does not appear to show a pronounced difference in the localization of mEos4b-GlyRβ on Sylite puncta between these two regions. If the images presented in Figure 3A are not representative, the authors should consider replacing them with more representative examples or providing an expanded images with multiple representative examples. Alternatively, if this inconsistency can be explained by differences in spot density within clusters, the authors should explain that.

(3) Quantification in Figure 5. It is recommended that the authors provide quantitative data on cluster formation and colocalization with Sylite puncta in Figure 5 to support their qualitative observations.

(4) Potential for pseudo replication. It's not clear whether they're performing stats tests across biological replica, images, or even synapses. They often quote mean +/- SEM with n = 1000s, and so does that mean they're doing tests on those 1000s? Need to clarify.

(5) Does mEoS effect expression levels or function of the protein? Can't see any experiments done to confirm this. Could suggest WB on homogenate, or mass spec?

(6) Quantification of protein numbers is challenging with SMLM. Issues include (i) some of FP not correctly folded/mature, and (ii) dependence of localisation rate on instrument, excitation/illumination intensities, and also the thresholds used in analysis. Can the authors compare with another protein that has known expression levels- e.g. PSD95? This is quite an ask, but if they could show copy number of something known to compare with, it would be useful.

(7) Rationale for doing nanobody dSTORM not clear at all. They don't explain the reason for doing the dSTORM experiments. Why not just rely on PALM for coincidence measurements, rather than tagging mEoS with a nanobody, and then doing dSTORM with that? Can they explain? Is it to get extra localisations- i.e. multiple per nanobody? If so, localising same FP multiple times wouldn't improve resolution. Also, no controls for nanobody dSTORM experiments- what about non-spec nb, or use on WT sections?

(8) What resolutions/precisions were obtained in SMLM experiments? Should perform Fourier Ring Correlation (FRC) on SR images to state resolutions obtained (particularly useful for when they're presenting distance histograms, as this will be dependent on resolution). Likewise for precision, what was mean precision? Can they show histograms of localisation precision.

(9) Why were DBSCAN parameters selected? How can they rule out multiple localisations per fluor? If low copy numbers (<10), then why bother with DBSCAN? Could just measure distance to each one.

(10) For microscopy experiment methods, state power densities, not % or "nominal power".

(11) In general, not much data presented. Any SI file with extra images etc.?

(12) Clarification of the discussion on GlyR expression and synaptic localization: The discussion on GlyR expression, complex formation, and synaptic localization is sometimes unclear, and needs terminological distinctions between "expression level", "complex formation" and "synaptic localization". For example, the authors state: "What then is the reason for the low protein expression of GlyRβ? One possibility is that the assembly of mature heteropentameric GlyR complexes depends critically on the expression of endogenous GlyR α subunits." Does this mean that GlyRβ proteins that fail to form complexes with GlyRα subunits are unstable and subject to rapid degradation? If so, the authors should clarify this point. The statement "This raises the interesting possibility that synaptic GlyRs may depend specifically on the concomitant expression of both α1 and β transcripts." suggests a dependency on α1 and β transcripts. However, is the authors' focus on synaptic localization or overall protein expression levels? If this means synaptic localization, it would be beneficial to state this explicitly to avoid confusion. To improve clarity, the authors should carefully distinguish between these different aspects of GlyR biology throughout the discussion. Additionally, a schematic diagram illustrating these processes would be highly beneficial for readers.

(13) Interpretation of GlyR localization in the context of nanodomains. The distribution of GlyR molecules on inhibitory synapses appears to be non-homogeneous, instead forming nanoclusters or nanodomains, similar to many other synaptic proteins. It is important to interpret GlyR localization in the context of nanodomain organization.

Significance:

The paper presents biological and technical advances. The biological insights revolve mostly on the documentation of Glycine receptors in particular synapses in forebrain, where they are typically expressed at very low levels. The authors provide compelling data indicating that the expression is of physiological significance. The authors have done a nice job of combining genetically tagged mice with advanced microscopy methods to tackle the question of distributions of synaptic proteins. Overall, these advances are more incremental than groundbreaking.

Comments on revised version:

The authors have addressed the majority of the significant issues raised in the review and revised the manuscript appropriately. One issue that can be further addressed relates to the issue of pseudo-replication. The authors state in their response that "All experiments were repeated at least twice to ensure reproducibility (N independent experiments). Statistical tests were performed on pooled data across the biological replicates; n denotes the number of data points used for testing (e.g., number of synaptic clusters, detections, cells, as specified in each case).". This suggests that they're not doing their stats on biological replicates, and instead are pseudo replicating. It's not clear how they have ensured reproducibility, when the stats seem to have been done on pooled data across repeats.

---

## [Referee Report · Reviewer #2 (Public review)]

Summary:

In their manuscript "Single molecule counting detects low-copy glycine receptors in hippocampal and striatal synapses" Camuso and colleagues apply single molecule localization microscopy (SMLM) methods to visualize low copy numbers of GlyRs at inhibitory synapses in the hippocampal formation and the striatum. SMLM analysis revealed higher copy numbers in striatum compared to hippocampal inhibitory synapses. They further provide evidence that these low copy numbers are tightly linked to post-synaptic scaffolding protein gephyrin at inhibitory synapses. Their approach profits from the high detection sensitivity and resolution of SMLM and challenges the controversial view on the presence of GlyRs in these formations although there are reports (electrophysiology) on the presence of GlyRs in these particular brain regions. These new datasets in the current manuscript may certainly assist in understanding the complexity of fundamental building blocks of inhibitory synapses.

Strengths:

The manuscript provides new insights to presence of low-copy numbers by visualizing them via SMLM. This is the first report that visualizes GlyR optically in the brain applying the knock-in model of mEOS4b tagged GlyRß and quantifies their copy number comparing distribution and amount of GlyRs from hippocampus and striatum. Imaging data correspond well to electrophysiological measurements in the manuscript.

Comments on revised version:

My concerns have been successfully addressed by the authors during the revision process.

---

## [Referee Report · Reviewer #3 (Public review)]

In this study, Camuso et al., make use of a knock-in mouse model expressing endogenously mEos4b-tagged GlyRβ subunits to detect endogenous glycine receptors in mouse brain using single-molecule localization microscopy (SMLM). At synapses in the hippocampus GlyRβ molecules are detected at very low copy numbers. Assuming that each detected GlyRβ molecule is incorporated in a pentameric glycine receptor, it was estimated that while the majority of hippocampal inhibitory synapses do not contain glycine receptors, a small population of inhibitory synapses contain a few (up to 10) glycine receptors. Using dual-color SMLM approaches it is furthermore confirmed that the detected GlyRβ molecules are embedded in the postsynaptic domain marked by gephyrin. In contrast to the hippocampus, at inhibitory synapses in the striatum GlyRβ molecules were detected at considerably higher copy numbers. Interestingly, the observed number of GlyRβ detections was significantly higher in the ventral striatum compared to the dorsal striatum. These findings are corroborated by electrophysiological recordings showing that postsynaptic glycinergic currents can be readily detected in the ventral striatum but are almost absent in the dorsal striatum. Using lentiviral overexpression of recombinant GlyRalpha1, alpha2, and beta subunits in cultured hippocampal neurons, it is shown that GlyR alpha1 subunits are readily detectable at synapses, but overexpressed GlyRalpha2 and beta subunits did not strongly enrich at synapses. This could indicate that GlyRa1 expression is limiting the synaptic expression of GlyRβ-containing glycine receptors in hippocampal neurons.

Comments on revised version:

This revised manuscript is significantly improved. New experimental and quantitative analysis is presented that strengthen the conclusions. Overall, the results presented in this manuscript are based on carefully performed SMLM experiments and are well-presented and described. The knock-in mouse with endogenously tagged GlyRβ molecules is a very strong aspect of this study and provides confidence in the labeling, the combination with SMLM is very strong as it provides high sensitivity and spatial resolution. These results confirm previous studies and will be of interest to a specialised audience interested in glycine receptors, inhibitory synapse biology and super-resolution microscopy.

---

## [Author Response]

The following is the authors’ response to the current reviews.

We thank the editors of eLife and the reviewers for their thorough evaluation of our study. As regards the final comments of reviewer 1 please note that all experimental replicates were first analyzed separately, and were then pooled, since the observed changes were comparable between experiments. This mean that statistical analyses were done on pooled biological replicates.

The following is the authors’ response to the original reviews.

**General Statements**

We thank the reviewers for their thorough and constructive evaluation of our work. We have revised the manuscript carefully and addressed all the criticisms raised, in particular the issues mentioned by several of the reviewers (see point-by-point response below). We have also added a number of explanations in the text for the sake of clarity, while trying to keep the manuscript as concise as possible.

In our view, the novelty of our research is two-fold. From a neurobiological point of view, we provide conclusive evidence for the existence of glycine receptors (GlyRs) at inhibitory synapses in various brain regions including the hippocampus, dentate gyrus and sub-regions of the striatum. This solves several open questions and has fundamental implications for our understanding of the organisation and function of inhibitory synapses in the telencephalon. Secondly, our study makes use of the unique sensitivity of single molecule localisation microscopy (SMLM) to identify low protein copy numbers. This is a new way to think about SMLM as it goes beyond a mere structural characterisation and towards a quantitative assessment of synaptic protein assemblies.

**Point-by-point description of the revisions**

**Reviewer #1 (Evidence, reproducibility and clarity):**
In this manuscript, the authors investigate the nanoscopic distribution of glycine receptor subunits in the hippocampus, dorsal striatum, and ventral striatum of the mouse brain using single-molecule localization microscopy (SMLM). They demonstrate that only a small number of glycine receptors are localized at hippocampal inhibitory synapses. Using dual-color SMLM, they further show that clusters of glycine receptors are predominantly localized within gephyrinpositive synapses. A comparison between the dorsal and ventral striatum reveals that the ventral striatum contains approximately eight times more glycine receptors and this finding is consistent with electrophysiological data on postsynaptic inhibitory currents. Finally, using cultured hippocampal neurons, they examine the differential synaptic localization of glycine receptor subunits (α1, α2, and β). This study is significant as it provides insights into the nanoscopic localization patterns of glycine receptors in brain regions where this protein is expressed at low levels. Additionally, the study demonstrates the different localization patterns of GlyR in distinct striatal regions and its physiological relevance using SMLM and electrophysiological experiments. However, several concerns should be addressed.The following are specific comments:(1) Colocalization analysis in Figure 1A. The colocalization between Sylite and mEos-GlyRβ appears to be quite low. It is essential to assess whether the observed colocalization is not due to random overlap. The authors should consider quantifying colocalization using statistical methods, such as a pixel shift analysis, to determine whether colocalization frequencies remain similar after artificially displacing one of the channels.

Following the suggestion of reviewer 1, we re-analysed CA3 images of Glrb^eos/eos^ hippocampal slices by applying a pixel-shift type of control, in which the Sylite channel (in far red) was horizontally flipped relative to the mEos4b-GlyRβ channel (in green, see Methods). As expected, the number of mEos4b-GlyRβ detections per gephyrin cluster was markedly reduced compared to the original analysis (revised Fig. 1B), confirming that the synaptic mEos4b detections exceed chance levels (see page 5).

(2) Inconsistency between Figure 3A and 3B. While Figure 3B indicates an ~8-fold difference in the number of mEos4b-GlyRβ detections per synapse between the dorsal and ventral striatum, Figure 3A does not appear to show a pronounced difference in the localization of mEos4bGlyRβ on Sylite puncta between these two regions. If the images presented in Figure 3A are not representative, the authors should consider replacing them with more representative examples or providing an expanded images with multiple representative examples. Alternatively, if this inconsistency can be explained by differences in spot density within clusters, the authors should explain that.

The pointillist images in Fig. 3A are essentially binary (red-black). Therefore, the density of detections at synapses cannot be easily judged by eye. For clarity, the original images in Fig. 3A have been replaced with two other examples that better reflect the different detection numbers in the dorsal and ventral striatum.

(3) Quantification in Figure 5. It is recommended that the authors provide quantitative data on cluster formation and colocalization with Sylite puncta in Figure 5 to support their qualitative observations.

This is an important point that was also raised by the other reviewers. We have performed additional experiments to increase the data volume for analysis. For quantification, we used two approaches. First, we counted the percentage of infected cells in which synaptic localisation of the recombinant receptor subunit was observed (Fig. 5C). We found that mEos4b-GlyRa1 consistently localises at synapses, indicating that all cells express endogenous GlyRb. When neurons were infected with mEos4b-GlyRb, fewer cells had synaptic clusters, meaning that indeed, GlyR alpha subunits are the limiting factor for synaptic targeting. In cultures infected with mEos4b-GlyRa2, only very few neurons displayed synaptic localisation (as judged by epifluorescence imaging). We think this shows that GlyRa2 is less capable of forming heteromeric complexes than GlyRa1, in line with our previous interpretation (see pp. 9-10, 13).

Secondly, we quantified the total intensity of each subunit at gephyrin-positive domains, both in infected neurons as well as non-infected control cultures (Fig. 5D). We observed that mEos4bGlyRa1 intensity at gephyrin puncta was higher than that of the other subunits, again pointing to efficient synaptic targeting of GlyRa1. Gephyrin cluster intensities (Sylite labelling) were not significantly different in GlyRb and GlyRa2 expressing neurons compared to the uninfected control, indicating that the lentiviral expression of recombinant subunits does not fundamentally alter the size of mixed inhibitory synapses in hippocampal neurons. Interestingly, gephyrin levels were slightly higher in hippocampal neurons expressing mEos4b-GlyRa1. In our view, this comes from an enhanced expression and synaptic targeting of mEos4b-GlyRa1 heteromers with endogenous GlyRb, pointing to a structural role of GlyRa1/b in hippocampal synapses (pp. 10, 13).

The new data and analyses have been described and illustrated in the relevant sections of the manuscript.

(4) Potential for pseudo replication. It's not clear whether they're performing stats tests across biological replica, images, or even synapses. They often quote mean +/- SEM with n = 1000s, and so does that mean they're doing tests on those 1000s? Need to clarify.

All experiments were repeated at least twice to ensure reproducibility (N independent experiments). Statistical tests were performed on pooled data across the biological replicates; n denotes the number of data points used for testing (e.g., number of synaptic clusters, detections, cells, as specified in each case). We have systematically given these numbers in the revised manuscript (n, N, and other experimental parameters such as the number of animals used, coverslips, images or cells). Data are generally given as mean +/- SEM or as mean +/- SD as indicated.

(5) Does mEoS effect expression levels or function of the protein? Can't see any experiments done to confirm this. Could suggest WB on homogenate, or mass spec?

The Glrb^eos/eos^ knock-in mouse line has been characterised previously and does not to display any ultrastructural or functional deficits at inhibitory synapses (Maynard et al. 2021 eLife). GlyRβ expression and glycine-evoked responses were not significantly different to those of the wildtype. The synaptic localisation of mEos4b-GlyRb in KI animals demonstrates correct assembly of heteromeric GlyRs and synaptic targeting. Accordingly, the animals do not display any obvious phenotype. We have clarified this in the manuscript (p. 4). In the case of cultured neurons, long-term expression of fluorescent receptor subunits with lentivirus has proven ideal to achieve efficient synaptic targeting. The low and continuous supply of recombinant receptors ensures assembly with endogenous subunits to form heteropentameric receptor complexes (e.g. [Patrizio et al. 2017 Sci Rep]). In the present study, lentivirus infection did not induce any obvious differences in the number or size of inhibitory synapses compared to control neurons, as judged by Sylite labelling of synaptic gephyrin puncta (new Fig. 5D).

(6) Quantification of protein numbers is challenging with SMLM. Issues include (i) some of FP not correctly folded/mature, and (ii) dependence of localisation rate on instrument, excitation/illumination intensities, and also the thresholds used in analysis. Can the authors compare with another protein that has known expression levels- e.g. PSD95? This is quite an ask, but if they could show copy number of something known to compare with, it would be useful.

We agree that absolute quantification with SMLM is challenging, since the number of detections depends on fluorophore maturation, photophysics, imaging conditions, and analysis thresholds (discussed in Patrizio & Specht 2016, Neurophotonics). For this reason, only very few datasets provide reliable copy numbers, even for well-studied proteins such as PSD-95. One notable exception is the study by Maynard et al. (eLife 2021) that quantified endogenous GlyRβcontaining receptors in spinal cord synapses using SMLM combined with correlative electron microscopy. The strength of this work was the use of a KI mouse strain, which ensures that mEos4b-GlyRβ expression follows intrinsic regional and temporal profiles. The authors reported a stereotypic density of ~2,000 GlyRs/µm² at synapses, corresponding to ~120 receptors per synapse in the dorsal horn and ~240 in the ventral horn, taking into account various parameters including receptor stoichiometry and the functionality of the fluorophore. These values are very close to our own calculations of GlyR numbers at spinal cord synapses that were obtained slightly differently in terms of sample preparation, microscope setup, imaging conditions, and data analysis, lending support to our experimental approach. Nevertheless, the obtained GlyR copy numbers at hippocampal synapses clearly have to be taken as estimates rather than precise figures, because the number of detections from a single mEos4b fluorophore can vary substantially, meaning that the fluorophores are not represented equally in pointillist images. This can affect the copy number calculation for a specific synapse, in particular when the numbers are low (e.g. in hippocampus), however, it should not alter the average number of detections (Fig. 1B) or the (median) molecule numbers of the entire population of synapses (Fig. 1C). We have discussed the limitations of our approach (p. 11).

(7) Rationale for doing nanobody dSTORM not clear at all. They don't explain the reason for doing the dSTORM experiments. Why not just rely on PALM for coincidence measurements, rather than tagging mEoS with a nanobody, and then doing dSTORM with that? Can they explain? Is it to get extra localisations- i.e. multiple per nanobody? If so, localising same FP multiple times wouldn't improve resolution. Also, no controls for nanobody dSTORM experiments- what about non-spec nb, or use on WT sections?

As discussed above (point 6), the detection of fluorophores with SMLM is influenced by many parameters, not least the noise produced by emitting molecules other than the fluorophore used for labelling. Our study is exceptional in that it attempts to identify extremely low molecule numbers (down to 1). To verify that the detections obtained with PALM correspond to mEos4b, we conducted robust control experiments (including pixel-shift as suggested by the reviewer, see point 1, revised Fig. 1B). The rationale for the nanobody-based dSTORM experiments was twofold: (1) to have an independent readout of the presence of low-copy GlyRs at inhibitory synapses and (2) to analyse the nanoscale organisation of GlyRs relative to the synaptic gephyrin scaffold using dual-colour dSTORM with spectral demixing (see p. 6). The organic fluorophores used in dSTORM (AF647, CF680) ensure high photon counts, essential for reliable co-localisation and distance analysis. PALM and dSTORM cannot be combined in dual-colour mode, as they require different buffers and imaging conditions.

The specificity of the anti-Eos nanobody was demonstrated by immunohistochemistry in spinal cord cultures expressing mEos4b-GlyRb and wildtype control tissue (Fig. S3). In response to the reviewer's remarks, we also performed a negative control experiment in Glrb^eos/eos^ slices (dSTORM), in which the nanobody was omitted (new Fig. S4F,G). Under these conditions, spectral demixing produced a single peak corresponding to CF680 (gephyrin) without any AF647 contribution (Fig. S4F). The background detection of "false" AF647 detections at synapses was significantly lower than in the slices labelled with the nanobody. We conclude that the fluorescence signal observed in our dual-colour dSTORM experiments arises from the specific detection of mEos4b-GlyRb by the nanobody, rather than from background, crossreactivity or wrong attribution of colour during spectral demixing. We have added these data and explanations in the results (p. 7) and in the figure legend of Fig. S4F,G.

(8) What resolutions/precisions were obtained in SMLM experiments? Should perform Fourier Ring Correlation (FRC) on SR images to state resolutions obtained (particularly useful for when they're presenting distance histograms, as this will be dependent on resolution). Likewise for precision, what was mean precision? Can they show histograms of localisation precision.

This is an interesting question in the context of our experiments with low-copy GlyRs, since the spatial resolution of SMLM is limited also by the density of molecules, i.e. the sampling of the structure in question (Nyquist-Shannon criterion). Accordingly, the priority of the PALM experiments was to improve the sensibility of SMLM for the identification of mEos4b-GlyRb subunits, rather than to maximize the spatial resolution. The mean localisation precision in PALM was 33 +/- 12 nm, as calculated from the fitting parameters of each detection (Zeiss, ZEN software), which ultimately result from their signal-to-noise ratio. This is a relatively low precision for SMLM, which can be explained by the low brightness of mEos4b compared to organic fluorophores together with the elevated fluorescence background in tissue slices.

In the case of dSTORM, the aim was to study the relative distribution of GlyRs within the synaptic scaffold, for which a higher localisation precision was required (p. 6). Therefore, detections with a precision ≥ 25 nm were filtered during analysis with NEO software (Abbelight). The retained detections had a mean localisation precision of 12 +/- 5 for CF680 (Sylite) and 11 +/- 4 for AF647 (nanobody). These values are given in the revised manuscript (pp. 18, 22).

(9) Why were DBSCAN parameters selected? How can they rule out multiple localisations per fluor? If low copy numbers (<10), then why bother with DBSCAN? Could just measure distance to each one.

Multiple detections of the same fluorophore are intrinsic to dSTORM imaging and have not been eliminated from the analysis. Small clusters of detections likely represent individual molecules (e.g. single receptors in the extrasynaptic regions, Fig. 2A). DBSCAN is a robust clustering method that is quite insensitive to minor changes in the choice of parameters. For dSTORM of synaptic gephyrin clusters (CF680), a relatively low length (80 nm radius) together with a high number of detections (≥ 50 neighbours) were chosen to reconstruct the postsynaptic domain with high spatial resolution (see point 8). In the case of the GlyR (nanobody-AF647), the clustering was done mostly for practical reasons, as it provided the coordinates of the centre of mass of the detections. The low stringency of this clustering (200 nm radius, ≥ 5 neighbours) effectively filters single detections that can result from background noise or incorrect demixing. An additional reference explaining the use of DBSCAN including the choice of parameters is given on p. 22 (see also R2 point 4).

(10) For microscopy experiment methods, state power densities, not % or "nominal power".

Done. We now report the irradiance (laser power density) instead of nominal power (pp. 18, 21).

(11) In general, not much data presented. Any SI file with extra images etc.?

The original submission included four supplementary figures with additional data and representative images that should have been available to the reviewer (Figs. S1-S4). The SI file has been updated during revision (new Fig. S4E-G).

(12) Clarification of the discussion on GlyR expression and synaptic localization: The discussion on GlyR expression, complex formation, and synaptic localization is sometimes unclear, and needs terminological distinctions between "expression level", "complex formation" and "synaptic localization". For example, the authors state:"What then is the reason for the low protein expression of GlyRβ? One possibility is that the assembly of mature heteropentameric GlyR complexes depends critically on the expression of endogenous GlyR α subunits." Does this mean that GlyRβ proteins that fail to form complexes with GlyRα subunits are unstable and subject to rapid degradation? If so, the authors should clarify this point. The statement "This raises the interesting possibility that synaptic GlyRs may depend specifically on the concomitant expression of both α1 and β transcripts." suggests a dependency on α1 and β transcripts. However, is the authors' focus on synaptic localization or overall protein expression levels? If this means synaptic localization, it would be beneficial to state this explicitly to avoid confusion. To improve clarity, the authors should carefully distinguish between these different aspects of GlyR biology throughout the discussion. Additionally, a schematic diagram illustrating these processes would be highly beneficial for readers.

We thank the reviewer to point this out. We are dealing with several processes; protein expression that determines subunit availability and the assembly of pentameric GlyRs complexes, surface expression, membrane diffusion and accumulation of GlyRb-containing receptor complexes at inhibitory synapses. We have edited the manuscript, particularly the discussion and tried to be as clear as possible in our wording.

We chose not to add a schematic illustration for the time being, because any graphical representation is necessarily a simplification. Instead, we preferred to summarise the main numbers in tabular form (Table 1). We are of course open to any other suggestions.

(13) Interpretation of GlyR localization in the context of nanodomains. The distribution of GlyR molecules on inhibitory synapses appears to be non-homogeneous, instead forming nanoclusters or nanodomains, similar to many other synaptic proteins. It is important to interpret GlyR localization in the context of nanodomain organization.

The dSTORM images in Fig. 2 are pointillist representations that show individual detections rather than molecules. Small clusters of detections are likely to originate from a single AF647 fluorophore (in the case of nanobody labelling) and therefore represent single GlyRb subunits. Since GlyR copy numbers are so low at hippocampal synapses (≤ 5), the notion of nanodomain is not directly applicable. Our analysis therefore focused on the integration of GlyRs within the postsynaptic scaffold, rather than attempting to define nanodomain structures (see also response to point 8 of R1). A clarification has been added in the revised manuscript (p. 6).

**Reviewer #1 (Significance):**
The paper presents biological and technical advances. The biological insights revolve mostly on the documentation of Glycine receptors in particular synapses in forebrain, where they are typically expressed at very low levels. The authors provide compelling data indicating that the expression is of physiological significance. The authors have done a nice job of combining genetically-tagged mice with advanced microscopy methods to tackle the question of distributions of synaptic proteins. Overall these advances are more incremental than groundbreaking.

We thank the reviewer for acknowledging both the technical and biological advances of our study. While we recognize that our work builds upon established models, we consider that it also addresses important unresolved questions, namely that GlyRs are present and specifically anchored at inhibitory synapses in telencephalic regions, such as the hippocampus and striatum. From a methodological point of view, our study demonstrates that SMLM can be applied not only for structural analysis of highly abundant proteins, but also to reliably detect proteins present at very low copy numbers. This ability to identify and quantify sparse molecule populations adds a new dimension to SMLM applications, which we believe increases the overall impact of our study beyond the field of synaptic neuroscience.

**Reviewer #2 (Evidence, reproducibility and clarity):**
In their manuscript "Single molecule counting detects low-copy glycine receptors in hippocampal and striatal synapses" Camuso and colleagues apply single molecule localization microscopy (SMLM) methods to visualize low copy numbers of GlyRs at inhibitory synapses in the hippocampal formation and the striatum. SMLM analysis revealed higher copy numbers in striatum compared to hippocampal inhibitory synapses. They further provide evidence that these low copy numbers are tightly linked to post-synaptic scaffolding protein gephyrin at inhibitory synapses. Their approach profits from the high sensitivity and resolution of SMLM and challenges the controversial view on the presence of GlyRs in these formations although there are reports (electrophysiology) on the presence of GlyRs in these particular brain regions. These new datasets in the current manuscript may certainly assist in understanding the complexity of fundamental building blocks of inhibitory synapses.However I have some minor points that the authors may address for clarification:(1) In Figure 1 the authors apply PALM imaging of mEos4b-GlyRß (knockin) and here the corresponding Sylite label seems to be recorded in widefield, it is not clearly stated in the figure legend if it is widefield or super-resolved. In Fig 1 A - is the scale bar 5 µm? Some Sylite spots appear to be sized around 1 µm, especially the brighter spots, but maybe this is due to the lower resolution of widefield imaging? Regarding the statistical comparison: what method was chosen to test for normality distribution, I think this point is missing in the methods section.

This is correct; the apparent size of the Sylite spots does not reflect the real size of the synaptic gephyrin domain due to the limited resolution of widefield imaging including the detection of outof-focus light. We have clarified in the legend of Fig. 1A that Sylite labelling was with classic epifluorescence microscopy. The scale bar in Fig. 1A corresponds to 5 µm. Since the data were not normally distributed, nonparametric tests (Kruskal- Wallis one-way ANOVA with Dunn’s multiple comparison test or Mann-Whitney U-test for pairwise comparisons) were used (p. 23).

Moreover I would appreciate a clarification and/or citation that the knockin model results in no structural and physiological changes at inhibitory synapses, I believe this model has been applied in previous studies and corresponding clarification can be provided.

The Glrbeos/eos mouse model has been described previously and does not exhibit any structural or physiological phenotypes (Maynard et al. 2021 eLife). The issue was also raised by reviewer R1 (point 5) and has been clarified in the revised manuscript (p. 4).

(2) In the next set of experiments the authors switch to demixing dSTORM experiments - an explanation why this is performed is missing in the text - I guess better resolution to perform more detailed distance measurements? For these experiments: which region of the hippocampus did the authors select, I cannot find this information in legend or main text.

Yes, the dSTORM experiments enable dual-colour structural analysis at high spatial resolution (see response to R1 point 7). An explanation has been added (p. 6).

(3) Regarding parameters of demixing experiments: the number of frames (10.000) seems quite low and the exposure time higher than expected for Alexa 647. Can the authors explain the reason for chosing these particular parameters (low expression profile of the target - so better separation?, less fluorophores on label and shorter collection time?) or is there a reference that can be cited? The laser power is given in the methods in percentage of maximal output power, but for better comparison and reproducibility I recommend to provide the values of a power meter (kW/cm2) as lasers may change their maximum output power during their lifetime.

Acquisition parameters (laser power, exposure time) for dSTORM were chosen to obtain a good localisation precision (~12 nm; see R1 point 8). The number of frames is adequate to obtain well sampled gephyrin scaffolds in the CF680 channel. In the case of the GlyR (nanobody-AF647), the concept of spatial resolution does not really apply due to the low number of targets (see R1, point 13). Power density (irradiance) values have now been given (pp. 18, 21).

(4) For analysis of subsynaptic distribution: how did the authors decide to choose the parameters in the NEO software for DBSCAN clustering - was a series of parameters tested to find optimal conditions and did the analysis start with an initial test if data is indeed clustered (K-ripley) or is there a reference in literature that can be provided?

DBSCAN parameters were optimised manually, by testing different values. Identification of dense and well-delimited gephyrin clusters (CF680) was achieved with a small radius and a high number of detections (80 nm, ≥ 50 neighbours), whereas filtering of low-density background in the AF647 channel (GlyRs) required less stringent parameters (200 nm, ≥ 5) due to the low number of target molecules. Similar parameters were used in a previous publication (Khayenko et al. 2022, Angewandte Chemie). The reference has been provided on p. 22 (see also R1 point 9).

(5) A conclusion/discussion of the results presented in Figure 5 is missing in the text/discussion.

This part of the manuscript has been completely overhauled. It includes new experimental data, quantification of the data (new Fig.5), as well as the discussion and interpretation of our findings (see also R1, point 3). In agreement with our earlier interpretation, the data confirm that low availability of GlyRa1 subunits limits the expression and synaptic targeting of GlyRa1/b heteropentamers. The observation that GlyRa1 overexpression with lentivirus increases the size of the postsynaptic gephyrin domain further points to a structural role, whereby GlyRs can enhance the stability (and size) of inhibitory synapses in hippocampal neurons, even at low copy numbers (pp. 13-14).

(6) In line 552 "suspension" is misleading, better use "solution"

Done.

**Reviewer #2 (Significance):**
Significance: The manuscript provides new insights to presence of low-copy numbers by visualizing them via SMLM. This is the first report that visualizes GlyR optically in the brain applying the knock-in model of mEOS4b tagged GlyRß and quantifies their copy number comparing distribution and amount of GlyRs from hippocampus and striatum. Imaging data correspond well to electrophysiological measurements in the manuscript.Field of expertise: Super-Resolution Imaging and corresponding analysis
**Reviewer #4 (Evidence, reproducibility and clarity):**
In this study, Camuso et al., make use of a knock-in mouse model expressing endogenously mEos4b-tagged GlyRβ to detect endogenous glycine receptors using single-molecule localization microscopy. The main conclusion from this study is that in the hippocampus GlyRβ molecules are barely detected, while inhibitory synapses in the ventral striatum seem to express functionally relevant GlyR numbers.I have a few points that I hope help to improve the strength of this study.- In the hippocampus, this study finds that the numbers of detections are very low. The authors perform adequate controls to indicate that these localizations are above noise level. Nevertheless, it remains questionable that these reflect proper GlyRs. The suggestion that in hippocampal synapses the low numbers of GlyRβ molecules "are important in assembly or maintenance of inhibitory synaptic structures in the brain" is on itself interesting, but is not at all supported. It is also difficult to envision how such low numbers could support the structure of a synapse. A functional experiment showing that knockdown of GlyRs affects inhibitory synapse structure in hippocampal neurons would be a minimal test of this.

It is not clear what the reviewer means by “it remains questionable that these reflect proper GlyRs”. The PALM experiments include a series of stringent controls (see R1, point 1) demonstrating the existence of low-copy GlyRs at inhibitory synapses in the hippocampus (Fig. 1) and in the striatum (Fig. 3), and are backed up by dSTORM experiments (Fig. 2). We have no reason to doubt that these receptors are fully functional as demonstrated for the ventral striatum (Fig. 4). However, due to their low number, a role in inhibitory synaptic transmission is clearly limited, at least in the hippocampus and dorsal striatum.

We therefore propose a structural role, where the GlyRs could be required to stabilise the postsynaptic gephyrin domain in hippocampal neurons. This is based on the idea that the GlyRgephyrin affinity is much higher than that of the GABAAR-gephyrin interaction (reviewed in Kasaragod & Schindelin 2018 Front Mol Neurosci). Accordingly, there is a close relationship between GlyRs and gephyrin numbers, sub-synaptic distribution, and dynamics in spinal cord synapses that are mostly glycinergic (Specht et al. 2013 Neuron; Maynard et al. 2021 eLife; Chapdelaine et al. 2021 Biophys J). It is reasonable to assume that low-copy GlyRs could play a similar structural role at hippocampal synapses. A knockdown experiment targeting these few receptors is technically very challenging and beyond the scope of this study. However, in response to the reviewer's question we have conducted new experiments in cultured hippocampal neurons (new Fig. 5). They demonstrate that overexpression of GlyRa1/b heteropentamers increases the size of the postsynaptic domain in these neurons, supporting our interpretation of a structural role of low-copy GlyRs (p. 14).

- The endogenous tagging strategy is a very strong aspect of this study and provides confidence in the labeling of GlyRβ molecules. One caveat however, is that this labeling strategy does not discriminate whether GlyRβ molecules are on the cell membrane or in internal compartments. Can the authors provide an estimate of the ratio of surface to internal GlyRβ molecules?

Gephyrin is known to form a two-dimensional scaffold below the synaptic membrane to which inhibitory GlyRs and GABAARs attach (reviewed in Alvarez 2017 Brain Res). The majority of the synaptic receptors are therefore thought to be located in the synaptic membrane, which is supported by the close relationship between the sub-synaptic distribution of GlyRs and gephyrin in spinal cord neurons (e.g. Maynard et al. 2021 eLife). To demonstrate the surface expression of GlyRs at hippocampal synapses we labelled cultured hippocampal neurons expressing mEos4b-GlyRa1 with anti-Eos nanobody in non-permeabilised neurons (see Author response image 1). The close correspondence between the nanobody (AF647) and the mEos4b signal confirms that the majority of the GlyRs are indeed located in the synaptic membrane.

**Author response image 1. sa4fig1:** Left: Lentivirus expression of mEos4b-GlyRa1 in fixed and non-permeabilised hippocampal neurons (mEos4b signal). Right: Surface labelling of the recombinant subunit with anti-Eos nanoboby (AF647).

- “We also estimated the absolute number of GlyRs per synapse in the hippocampus. The number of mEos4b detections was converted into copy numbers by dividing the detections at synapses by the average number of detections of individual mEos4b-GlyRβ containing receptor complexes”. In essence this is a correct method to estimate copy numbers, and the authors discuss some of the pitfalls associated with this approach (i.e., maturation of fluorophore and detection limit). Nevertheless, the authors did not subtract the number of background localizations determined in the two negative control groups. This is critical, particularly at these low-number estimations.

We fully agree that background subtraction can be useful with low detection numbers. In the revised manuscript, copy numbers are now reported as background-corrected values. Specifically, the mean number of detections measured in wildtype slices was used to calculate an equivalent receptor number, which was then subtracted from the copy number estimates across hippocampus, spinal cord and striatum. This procedure is described in the methods (p. 20) and results (p. 5, 8), and mentioned in the figure legends of Fig. 1C, 3C. The background corrected values are given in the text and Table 1.

- Furthermore, the authors state that "The advantage of this estimation is that it is independent of the stoichiometry of heteropentameric GlyRs". However, if the stoichometry is unknown, the number of counted GlyRβ subunits cannot simply be reported as the number of GlyRs. This should be discussed in more detail, and more carefully reported throughout the manuscript.

The reviewer is right to point this out. There is still some debate about the stoichiometry of heteropentameric GlyRs. Configurations with 2a:3b, 3a:2b and 4a:1b subunits have been advanced (e.g. Grudzinska et al. 2005 Neuron; Durisic et al. 2012 J Neurosci; Patrizio et al. 2017 Sci Rep; Zhu & Gouaux 2021 Nature). We have therefore chosen a quantification that is independent of the underlying stoichiometry. Since our quantification is based on very sparse clusters of mEos4b detections that likely originate from a single receptor complex (irrespective of its stoichiometry), the reported values actually reflect the number of GlyRs (and not GlyRb subunits). We have clarified this in the results (p. 5) and throughout the manuscript (Table 1).

- The dual-color imaging provides insights in the subsynaptic distribution of GlyRβ molecules in hippocampal synapses. Why are similar studies not performed on synapses in the ventral striatum where functionally relevant numbers of GlyRβ molecules are found? Here insights in the subsynaptic receptor distribution would be of much more interest as it can be tight to the function.

This is an interesting suggestion. However, the primary aim of our study was to identify the existence of GlyRs in hippocampal regions. At low copy numbers, the concept of sub-synaptic domains (SSDs, e.g. Yang et al. 2021 EMBO Rep) becomes irrelevant (see R1 point 13). It should be pointed out that the dSTORM pointillist images (Fig. 2A) represent individual GlyR detections rather than clusters of molecules. In the striatum, our specific purpose was to solve an open question about the presence of GlyRs in different subregions (putamen, nucleus accumbens).

- It is unclear how the experiments in Figure 5 add to this study. These results are valid, but do not seem to directly test the hypothesis that "the expression of α subunits may be limiting factor controlling the number of synaptic GlyRs". These experiments simply test if overexpressed α subunits can be detected. If the α subunits are limiting, measuring the effect of α subunit overexpression on GlyRβ surface expression would be a more direct test.

Both R1 and R2 have also commented on the data in Fig. 5 and their interpretation. We have substantially revised this section as described before (see R1 point 3) including additional experiments and quantification of the data (new Fig. 5). The findings lend support to our earlier hypothesis that GlyR alpha subunits (in particular GlyRa1) are the limiting factor for the expression of heteropentameric GlyRa/b in hippocampal neurons (pp. 13-14). Since the GlyRa1 subunit itself does not bind to gephyrin (Patrizio et al. 2017 Sci Rep), the synaptic localisation of the recombinant mEos4b-GlyRa1 subunits is proof that they have formed heteropentamers with endogenous GlyRb subunits and driven their membrane trafficking, which the GlyRb subunits are incapable of doing on their own.

**Reviewer #4 (Significance):**
These results are based on carefully performed single-molecule localization experiments, and are well-presented and described. The knockin mouse with endogenously tagged GlyRβ molecules is a very strong aspect of this study and provides confidence in the labeling, the combination with single-molecule localization microscopy is very strong as it provides high sensitivity and spatial resolution.The conceptual innovation however seems relatively modest, these results confirm previous studies but do not seem to add novel insights. This study is entirely descriptive and does not bring new mechanistic insights.This study could be of interest to a specialized audience interested in glycine receptor biology, inhibitory synapse biology and super-resolution microscopy.My expertise is in super-resolution microscopy, synaptic transmission and plasticity

As we have stated before, the novelty of our study lies in the use of SMLM for the identification of very small numbers of molecules, which requires careful control experiments. This is something that has not been done before and that can be of interest to a wider readership, as it opens up SMLM for ultrasensitive detection of rare molecular events. Using this approach, we solve two open scientific questions: (1) the demonstration that low-copy GlyRs are present at inhibitory synapses in the hippocampus, (2) the sub-region specific expression and functional role of GlyRs in the ventral versus dorsal striatum.

The following review was provided later under the name “Reviewer #4”. To avoid confusion with the last reviewer from above we will refer to this review as R4-2.

**Reviewer #4-2 (Evidence, reproducibility and clarity):**
Summary:Provide a short summary of the findings and key conclusions (including methodology and model system(s) where appropriate).The authors investigate the presence of synaptic glycine receptors in the telencephalon, whose presence and function is poorly understood.Using a transgenically labeled glycine receptor beta subunit (Glrb-mEos4b) mouse model together with super-resolution microscopy (SLMM, dSTORM), they demonstrate the presence of a low but detectable amount of synaptically localized GLRB in the hippocampus. While they do not perform a functional analysis of these receptors, they do demonstrate that these subunits are integrated into the inhibitory postsynaptic density (iPSD) as labeled by the scaffold protein gephyrin. These findings demonstrate that a low level of synaptically localized glycerine receptor subunits exist in the hippocampal formation, although whether or not they have a functional relevance remains unknown.They then proceed to quantify synaptic glycine receptors in the striatum, demonstrating that the ventral striatum has a significantly higher amount of GLRB co-localized with gephyrin than the dorsal striatum or the hippocampus. They then recorded pharmacologically isolated glycinergic miniature inhibitory postsynaptic currents (mIPSCs) from striatal neurons. In line with their structural observations, these recordings confirmed the presence of synaptic glycinergic signaling in the ventral striatum, and an almost complete absence in the dorsal striatum. Together, these findings demonstrate that synaptic glycine receptors in the ventral striatum are present and functional, while an important contribution to dorsal striatal activity is less likely.Lastly, the authors use existing mRNA and protein datasets to show that the expression level of GLRA1 across the brain positively correlates with the presence of synaptic GLRB.The authors use lentiviral expression of mEos4b-tagged glycine receptor alpha1, alpha2, and beta subunits (GLRA1, GLRA1, GLRB) in cultured hippocampal neurons to investigate the ability of these subunits to cause the synaptic localization of glycine receptors. They suggest that the alpha1 subunit has a higher propensity to localize at the inhibitory postsynapse (labeled via gephyrin) than the alpha2 or beta subunits, and may therefore contribute to the distribution of functional synaptic glycine receptors across the brain.Major comments:- Are the key conclusions convincing?The authors are generally precise in the formulation of their conclusions.(1) They demonstrate a very low, but detectable, amount of a synaptically localized glycine receptor subunit in a transgenic (GlrB-mEos4b) mouse model. They demonstrate that the GLRB-mEos4b fusion protein is integrated into the iPSD as determined by gephyrin labelling. The authors do not perform functional tests of these receptors and do not state any such conclusions.(2) The authors show that GLRB-mEos4b is clearly detectable in the striatum and integrated into gephyrin clusters at a significantly higher rate in the ventral striatum compared to the dorsal striatum, which is in line with previous studies.(3) Adding to their quantification of GLRB-mEos4b in the striatum, the authors demonstrate the presence of glycinergic miniature IPSCs in the ventral striatum, and an almost complete absence of mIPSCs in the dorsal striatum. These currents support the observation that GLRB-mEos4b is more synaptically integrated in the ventral striatum compared to the dorsal striatum.(4) The authors show that lentiviral expression of GLRA1-mEos4b leads to a visually higher number of GLR clusters in cultured hippocampal neurons, and a co-localization of some clusters with gephyrin. The authors claim that this supports the idea that GLRA1 may be an important driver of synaptic glycine receptor localization. However, no quantification or statistical analysis of the number of puncta or their colocalization with gephyrin is provided for any of the expressed subunits. Such a claim should be supported by quantification and statistics

A thorough analysis and quantification of the data in Fig.5 has been carried out as requested by all the other reviewers (e.g. R1, point 3). The new data and results have been described in the revised manuscript (pp. 9-10, 13-14).

- Should the authors qualify some of their claims as preliminary or speculative, or remove them altogether?One unaddressed caveat is the fact that a GLRB-mEos4b fusion protein may behave differently in terms of localization and synaptic integration than wild-type GLRB. While unlikely, it is possible that mEos4b interacts either with itself or synaptic proteins in a way that changes the fused GLRB subunit’s localization. Such an effect would be unlikely to affect synaptic function in a measurable way, but might be detected at a structural level by highly sensitive methods such as SMLM and STORM in regions with very low molecule numbers (such as the hippocampus). Since reliable antibodies against GLRB in brain tissue sections are not available, this would be difficult to test. Considering that no functional measures of the hippocampal detections exist, we would suggest that this possible caveat be mentioned for this particular experiment.

This question has also been raised before (R1, point 5). According to an earlier study the mEos4b-GlyRb knock-in does not cause any obvious phenotypes, with the possible exception of minor loss of glycine potency (Maynard et al. 2021 eLife). The fact that the synaptic levels in the spinal cord in heterozygous animals are precisely half of those of homozygous animals argues against differences in receptor expression, heteropentameric assembly, forward trafficking to the plasma membrane and integration into the synaptic membrane as confirmed using quantitative super-resolution CLEM (Maynard et al. 2021 eLife). Accordingly, we did not observe any behavioural deficits in these animals, making it a powerful experimental model. We have added this information in the revised manuscript (p. 4).

In addition, without any quantification or statistical analysis, the author’s claims regarding the necessity of GLRA1 expression for the synaptic localization of glycine receptors in cultured hippocampal neurons should probably be described as preliminary (Fig. 5).

As mentioned before, we have substantially revised this part (R1, point 3). The quantification and analysis in the new Fig. 5 support our earlier interpretation.

- Would additional experiments be essential to support the claims of the paper? Request additional experiments only where necessary for the paper as it is, and do not ask authors to open new lines of experimentation.The authors show that there is colocalization of gephyrin with the mEos4b-GlyRβ subunit using the Dual-colour SMLM. This is a powerful approach that allows for a claim to be made on the synaptic location of the glycine receptors. The images presented in Figure 1, together with the distance analysis in Figure 2, display the co-localization of the fluorophores. The co-localization images in all the selected regions, hippocampus and striatum, also show detections outside of the gephyrin clusters, which the authors refer to as extrasynaptic. These punctated small clusters seem to have the same size as the ones detected and assigned as part of the synapse. It would be informative if the authors analysed the distribution, density and size of these nonsynaptic clusters and presented the data in the manuscript and also compared it against the synaptic ones. Validating this extrasynaptic signal by staining for a dendritic marker, such as MAP-2 or maybe a somatic marker and assessing the co-localization with the non-synaptic clusters would also add even more credibility to them being extrasynaptic.

The existence of extrasynaptic GlyRs is well attested in spinal cord neurons (e.g. Specht et al. 2013 Neuron; this study see Fig. S2). The fact that these appear as small clusters of detections in SMLM recordings results from the fact that a single fluorophore can be detected several times in consecutive image frames and because of blinking. Therefore, small clusters of detections likely represent single GlyRs (that can be counted), and not assemblies of several receptor complexes. Due to their diffusion in the neuronal membrane, they are seen as diffuse signals throughout the somatodendritic compartment in epifluorescence images (e.g. Fig. 5A). SMLM recordings of the same cells resolves this diffuse signal into discrete nanoclusters representing individual receptors (Fig. 5B). It is not clear what information co-localisation experiments with specific markers could provide, especially in hippocampal neurons, in which the copy numbers (and density) of GlyRs is next to zero.

In addition we would encourage the authors to quantify the clustering and co-localization of virally expressed GLRA1, GLRA2, and GLRB with gephyrin in order to support the associated claims (Fig. 5). Preferably, the density of GLR and gephyrin clusters (at least on the somatic surface, the proximal dendrites, or both) as well as their co-localization probability should be quantified if a causal claim about subunit-specific requirements for synaptic localization is to be made.

Quantification of the data have been carried out (new Fig.5C,D). The results have been described before (R1, point 3) and support our earlier interpretation of the data (pp. 13-14).

Lastly, even though it may be outside of the scope of such a study analysing other parts of the hippocampal area could provide additional important information. If one looks at the Allen Institute’s ISH of the beta subunit the strongest signal comes from the stratum oriens in the CA1 for example, suggesting that interneurons residing there would more likely have a higher expression of the glycine receptors. This could also be assessed by looking more carefully at the single cell transcriptomics, to see which cell types in the hippocampus show the highest mRNA levels. If the authors think that this is too much additional work, then perhaps a mention of this in the discussion would be good.

We have added the requested information from the ISH database of the Allen Institute in the discussion as suggested by the reviewer (p. 12). However, in combination with the transcriptomic data (Fig. S1) our finding strongly suggest that the expression of synaptic GlyRs depends on the availability of alpha subunits rather than on the presence of the GlyRb transcript. This is obvious when one compares the mRNA levels in the hippocampus with those in the basal ganglia (striatum) and medulla. While the transcript concentrations of GlyRb are elevated in all three regions and essentially the same, our data show that the GlyRb copy numbers at synapses differ over more than 2 orders of magnitude (Fig. 1B, Table 1).

- Are the suggested experiments realistic in terms of time and resources? It would help if you could add an estimated cost and time investment for substantial experiments.Since the labeling and some imaging has been performed already, the requested experiment would be a matter of deploying a method of quantification. In principle, it should not require any additional wet-lab experiments, although it may require additional imaging of existing samples.- Are the data and the methods presented in such a way that they can be reproduced?Yes, for the most part.- Are the experiments adequately replicated and statistical analysis adequate?YesMinor comments:- Specific experimental issues that are easily addressable.N/A- Are prior studies referenced appropriately?Yes- Are the text and figures clear and accurate?Yes, although quantification in figure 5 is currently not present.

A quantification has been added (see R1, point 3).

- Do you have suggestions that would help the authors improve the presentation of their data and conclusions?This paper presents a method that could be used to localize receptors and perhaps other proteins that are in low abundance or for which a detailed quantification is necessary. I would therefore suggest that Figure S4 is included into Figure 2 as the first panel, showcasing the demixing, followed by the results.

We agree in principle with this suggestion. However, the revised Fig. S4 is more complex and we think that it would distract from the data shown in Fig. 2. Given that Fig. S4 is mostly methodological and not essential to understand the text, we have kept it in the supplement for the time being. We leave the final decision on this point to the editor.

**Reviewer #4-2 (Significance):**
[This review was supplied later]- Describe the nature and significance of the advance (e.g. conceptual, technical, clinical) for the field.Using a novel and high resolution method, the authors have provided strong evidence for the presence of glycine receptors in the murine hippocampus and in the dorsal striatum. The number of receptors calculated is small compared to the numbers found in the ventral striatum. This is the first study to quantify receptor numbers in these region. In addition it also lays a roadmap for future studies addressing similar questions.- Place the work in the context of the existing literature (provide references, where appropriate).This is done well by the authors in the curation of the literature. As stated above, the authors have filled a gap in the presence of glycine receptors in different brain regions, a subject of importance in understanding the role they play in brain activity and function.- State what audience might be interested in and influenced by the reported findings.Neuroscientists working at the synaptic level, on inhibitory neurotransmission and on fundamental mechanisms of expression of genes at low levels and their relationship to the presence of the protein would be interested. Furthermore, researchers in neuroscience and cell biology may benefit from and be inspired by the approach used in this manuscript, to potentially apply it to address their own aims.

We thank the reviewer for the positive assessment of the technical and biological implications of our work, as well as the interest of our findings to a wide readership of neuroscientists and cell biologists.

- Define your field of expertise with a few keywords to help the authors contextualize your point of view. Indicate if there are any parts of the paper that you do not have sufficient expertise to evaluate.Synaptic transmission, inhibitory cells and GABAergic synapses functionally and structurally, cortex and cortical circuits. No strong expertise in super-resolution imaging methods.